# Altered conformational sampling along an evolutionary trajectory changes the catalytic activity of an enzyme

Joe A. Kaczmarski[1,7], Mithun C. Mahawaththa[1,7], Akiva Feintuch[2,7], Ben E. Clifton[1,6], Luke A. Adams [3], Daniella Goldfarb [2,8 ✉], Gottfried Otting [1,4,8 ✉] & Colin J. Jackson [1,4,5,8 ✉]

Several enzymes are known to have evolved from non-catalytic proteins such as solute-binding proteins (SBPs). Although attention has been focused on how a binding site can evolve to become catalytic, an equally important question is: how do the structural dynamics of a binding protein change as it becomes an efficient enzyme? Here we performed a variety of experiments, including propargyl-DO3A-Gd(III) tagging and double electron–electron resonance (DEER) to study the rigid body protein dynamics of reconstructed evolutionary intermediates to determine how the conformational sampling of a protein changes along an evolutionary trajectory linking an arginine SBP to a cyclohexadienyl dehydratase (CDT). We observed that primitive dehydratases predominantly populate catalytically unproductive conformations that are vestiges of their ancestral SBP function. Non-productive conformational states, including a wide-open state, are frozen out of the conformational landscape via remote mutations, eventually leading to extant CDT that exclusively samples catalytically relevant compact states. These results show that remote mutations can reshape the global conformational landscape of an enzyme as a mechanism for increasing catalytic activity.

[1] Research School of Chemistry, The Australian National University, Canberra, ACT 2601, Australia. [2] Department of Chemical and Biological Physics, Weizmann Institute of Science, Rehovot 76100, Israel. [3] Medicinal Chemistry, Monash Institute of Pharmaceutical Sciences, Monash University, Parkville, VIC 3052, Australia. [4] Australian Research Council Centre of Excellence for Innovations in Peptide and Protein Science, Research School of Chemistry, Australian National University, Canberra 2601 ACT, Australia. [5] Australian Research Council Centre of Excellence in Synthetic Biology, Research School of Chemistry, Australian National University, Canberra 2601 ACT, Australia. [6]Present address: Protein Engineering and Evolution Unit, Okinawa Institute of Science and Technology, 1919-1 Tancha, Onna-son, Okinawa 904-0412, Japan. [7]These authors contributed equally: Joe A. Kaczmarski, Mithun C. Mahawaththa, Akiva Feintuch. [8]These authors jointly supervised this work: Daniella Goldfarb, Gottfried Otting, Colin J. Jackson. ✉email: daniella.goldfarb@weizmann.ac.il; gottfried.otting@anu.edu.au; colin.jackson@anu.edu.au

Solute-binding proteins (SBPs) comprise a large superfamily of extra-cytoplasmic receptors that are predominantly involved in sensing and the uptake of nutrients, including amino acids, carbohydrates, vitamins, metals and osmolytes[1–4]. The SBP fold consists of two α/β domains linked by a flexible hinge region that mediates a conformational change in solution, with hinge-bending and hinge-twisting motions moving the two domains together (closed state) and apart (open state; Fig. 1a). Ligands bind at the cleft between the two domains and stabilise the closed state by forming bridging interactions between the two domains and the hinge region[1]. The ligand-induced open-to-closed conformational switch is important for the function of SBPs in ATP-binding cassette transporter systems[1,5,6], tripartite ATP-independent periplasmic transporter systems[7] and signalling cascades[4].

Although the open conformation is the ground state for most ligand-free SBPs[1,5,8–19], numerous SBPs also sample semi-closed and closed states in the absence of ligands. For example, X-ray crystallography, nuclear magnetic resonance (NMR), molecular dynamics (MD) simulation, double electron–electron resonance (DEER, a.k.a. PELDOR) and Förster resonance energy transfer (FRET) studies on the maltose-binding protein (MBP)[20–22], glucose–galactose-binding protein[23,24], histidine-binding protein[25], ferri-bacillibactin-binding protein[26], glutamine-binding protein[27,28] and choline/acetylcholine-binding protein[29] demonstrate a sampling of both open and closed conformations in the absence of ligands. The intrinsic open/closed equilibrium is fundamental to determining the binding affinity[21,23,30,31] and binding promiscuity[5,32] of SBPs, and controlling the transport activity of SBP-associated systems[5,28]. Indeed, the extent of the open/closed motion differs between SBPs[1], and the function of an SBP can be changed by mutations that alter conformational sampling, without changing the architecture of the SBP-ligand interface[15,21,30].

While most SBPs are non-catalytic binding proteins, a small fraction of proteins within the periplasmic binding protein-fold superfamily have evolved enzymatic activity[33–37]. One example is cyclohexadienyl dehydratase (CDT), which is closely related to the polar and cationic amino acid-binding proteins, such as the arginine-binding protein (ArgBP)[37,38]. Like ArgBP, CDT adopts a periplasmic type-II SBP fold, but instead of binding amino acids, it catalyzes the Grob-like fragmentation of prephenate and L-arogenate to form phenylpyruvate and L-phenylalanine, respectively[39]. We previously used ancestral sequence reconstruction to show that the trimeric CDT from *Pseudomonas aeruginosa* (*Pa*CDT) plausibly evolved from a monomeric cationic amino acid-binding protein ancestor (AncCDT-1) via a series of intermediates (AncCDT-2 to AncCDT-5)[40] (Fig. 1b). While AncCDT-1 and AncCDT-2 did not display any enzymatic activity and appeared to be binding proteins, low-level CDT activity became detectable in two alternative reconstructions of AncCDT-3(P188/L188) ($k_{cat}/K_M$ ~$10^1$/$10^2$ s$^{-1}$ M$^{-1}$) and increased along the trajectory towards the efficient extant enzyme *Pa*CDT ($k_{cat}/K_M$ ~$10^6$ s$^{-1}$ M$^{-1}$). Despite the large difference in catalytic efficiency,

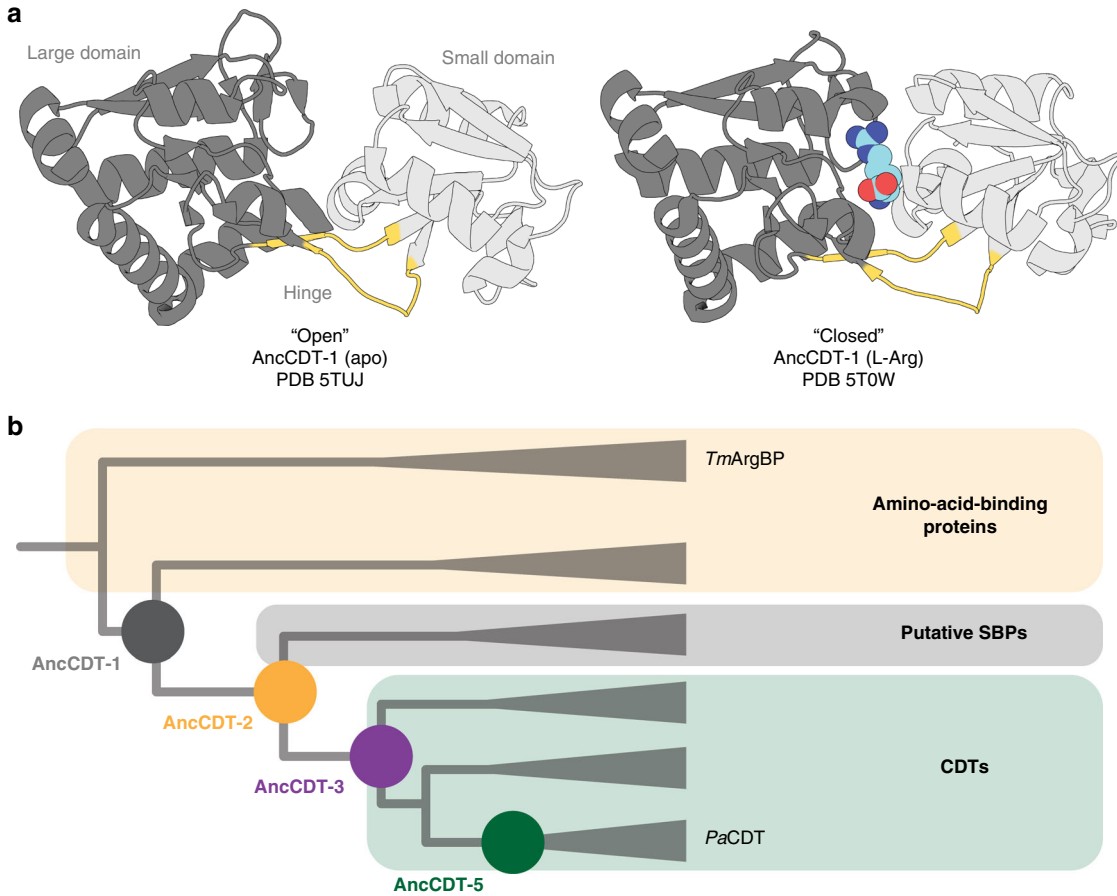

**Fig. 1 The conformational change of SBPs and the evolution of *Pa*CDT. a** X-ray crystal structures of SBPs that are specialised for binding solutes, such as AncCDT-1 (shown), typically capture open ligand-free (left, PDB 5TUJ) and closed liganded (right, PDB 5T0W) states. **b** Schematic drawing (not to scale) of the phylogenetic tree used for ancestral sequence reconstruction in Clifton et al.[40], which highlights the evolutionary relationship between the polar amino acid-binding proteins (e.g., *Thermotoga maritima* L-arginine-binding protein, *Tm*ArgBP), AncCDT-1, AncCDT-3, AncCDT-5 and *Pa*CDT. Clades are collapsed. Figure adapted from Clifton et al.[40].

**Table 1 Kinetic parameters for prephenate dehydratase activity of CDT variants related to this study[a].**

| Protein | $K_M$ (µM) | $k_{cat}$ (s$^{-1}$) | $k_{cat}/K_M$ (M$^{-1}$ s$^{-1}$) |
|---|---|---|---|
| AncCDT-1[40] | ND | ND | ND |
| AncCDT-3/P188[40] | 1830 ± 190 | $(1.04 ± 0.07) × 10^{-2}$ | 5.67 ± 0.70 |
| AncCDT-3/L188[40] | 294 ± 27 | $(4.58 ± 0.30) × 10^{-2}$ | 155 ± 18 |
| AncCDT-5 | 277 ± 4 | 4.1 ± 0.2 | $(1.5 ± 0.1) × 10^4$ |
| PaCDT[40] | 18.7 ± 2.9 | 18.4 ± 0.7 | $(9.8 ± 1.6) × 10^5$ |

ND not detected.
[a]The results are reported as mean ± standard error for $K_M$ and $k_{cat}$. Errors for $k_{cat}/K_M$ were obtained by propagation. Source data are provided as a Source Data file.

AncCDT-3 and PaCDT share 14/15 inner-shell residues, including all catalytic residues, but only about 50% amino acid sequence identity over the rest of the protein (the outer shells). This suggests that these outer-shell substitutions must be substantially responsible for the ~10$^5$-fold increase in catalytic efficiency. For example, the remote P188L substitution in AncCDT-3 was shown to result in a 27-fold increase in $k_{cat}/K_M$[40].

In this study, we investigated two plausible explanations for how remote mutations could have led to an increase in catalytic activity along this evolutionary trajectory: they may have (i) altered the sampling of rotamers of active site residues, such as the general acid Glu173, thereby changing the structure and character of the active site and controlling the configuration of active site residues, and/or (ii) altered the equilibrium between open and closed states of the protein to minimise sampling of the catalytically unproductive open state. To test these hypotheses, we used a combination of protein crystallography, MD simulations and DEER distance measurements on protein variants into which we incorporated the unnatural amino acid *p*-azidophenylalanine (AzF) to allow biorthogonal conjugation, making it specific even in the presence of native cysteine residues, to a propargyl-DO3A-Gd(III) tag, which encapsulates the Gd(III) ion in a hydrophilic complex of neutral charge, while providing a relatively rigid linker between the Gd(III) ion and protein backbone[41]. The Gd(III)–Gd(III) DEER measurements were carried out at W-band frequency (94.9 GHz), affording superior sensitivity and measurements free from orientation selection[42,43] and multi-spin effects[44–47]. The results from these experiments showed that the ancestral proteins, which all displayed either no catalytic activity (AncCDT-1) or very high $K_M$ values that were outside the physiologically relevant substrate concentration (AncCDT-3, AncCDT-5), significantly or even predominantly sampled open states, including a wide-open state that is not observed in the extant and efficient enzyme PaCDT. Finally, the structural analysis revealed that multiple changes to intra- and intermolecular (via oligomerization) interaction networks have shifted the conformational equilibrium towards more compact states along the evolutionary trajectory.

## Results

**The evolution of prephenate dehydratase activity.** We previously showed that while AncCDT-1 binds L-arginine with comparable affinity to extant L-arginine-binding proteins, the P188 AncCDT-3 reconstruction (hereafter AncCDT-3/P188), AncCDT-5 and PaCDT have all lost the ancestral ability to bind proteogenic amino acids[40]. Instead, along with the AncCDT-3/L188 AncCDT-3 reconstruction, they can rescue the growth of a phenylalanine auxotroph of *E. coli* (ΔpheA) grown in the absence of L-phenylalanine, indicating their capacity to deliver the dehydratase activity required to synthesise L-phenylalanine from L-arogenate, or the precursor phenylpyruvate from prephenate[40]. Furthermore, these variants exhibited prephenate dehydratase activity in vitro, which increased about 10$^5$-fold ($k_{cat}/K_M$) between AncCDT-3/P188 and PaCDT. In this work, we have

additionally assessed the dehydratase activity of AncCDT-5, which proved to be an intermediate between the AncCDT-3 variants and the extant enzyme PaCDT (Table 1 and Supplementary Fig. 1). Although the $k_{cat}$ value of AncCDT-5 (4 s$^{-1}$) exceeds the level observed in the AncCDT-3 variants (~10$^{-2}$ s$^{-1}$) almost as much as PaCDT (18 s$^{-1}$), the $K_M$ of AncCDT-5 remains ~15-fold higher than that of PaCDT (277 µM vs. 19 µM). This value is well above the likely physiological concentration of the substrate (the intracellular concentrations of molecules in this biosynthetic pathway are ~14–18 µM in Gram-negative bacteria[48]).

**Remote mutations affect active site configurations and catalysis.** Sequence and structural analysis of AncCDT-3/L188 and PaCDT revealed that although their catalytic efficiency differs by ~10$^5$ M$^{-1}$ s$^{-1}$, they share 14 of 15 residues in the active/substrate-binding site (the only difference being a conservative Thr80Ser substitution)[40]. Comparison between the structures of AncCDT-3/L188 and PaCDT revealed that the catalytic general acid Glu173 adopts a rotameric state in AncCDT-3/L188 that differs from that observed in PaCDT (Fig. 2a)[40]. This conformational difference appears to be caused by the neighbouring Tyr177Gln mutation that occurs on the branch between AncCDT-3 and AncCDT-5; the presence of Tyr177 in AncCDT-3 prevents Glu173 from adopting the catalytically competent conformation observed in PaCDT[40]. Additionally, this mutation causes coupled conformational changes in the neighbouring residues Phe156 and Met167, which leads to further changes in the shape and electrostatic character of the active site. These findings are analogous to those from recent work on enzyme variants produced through directed evolution and computational design that show how second-shell mutations affect enzyme activity by constraining the conformational sampling of catalytic residues[49,50].

To better understand how the mutations along this evolutionary trajectory affect activity, we also solved a 1.49 Å X-ray crystal structure of AncCDT-5 (Supplementary Table 1), which shares 15/15 residues in the active/substrate-binding site with PaCDT. In contrast to the comparison between AncCDT-3 and PaCDT, a structural alignment of AncCDT-5 and PaCDT (PDB 3KBR) reveals that these two proteins not only share identical binding sites in terms of amino acid composition but that these residues also adopt identical conformations (Fig. 2b). Since these active/binding sites are identical, the ~15-fold decrease in $K_M$ and ~5-fold increase in $k_{cat}$ of the dehydratase activity must be attributed to amino acid substitutions that are outside of the binding site. The sequences of AncCDT-5 and PaCDT differ at 98 positions. Excluding the nine amino acid C-terminal extension of PaCDT, the remaining 89 positions are evenly distributed around the SBP fold (Fig. 2c), making it difficult to rationalise how individual mutations contribute to changes in catalytic profiles. Instead, it is possible that some of the substitutions may be

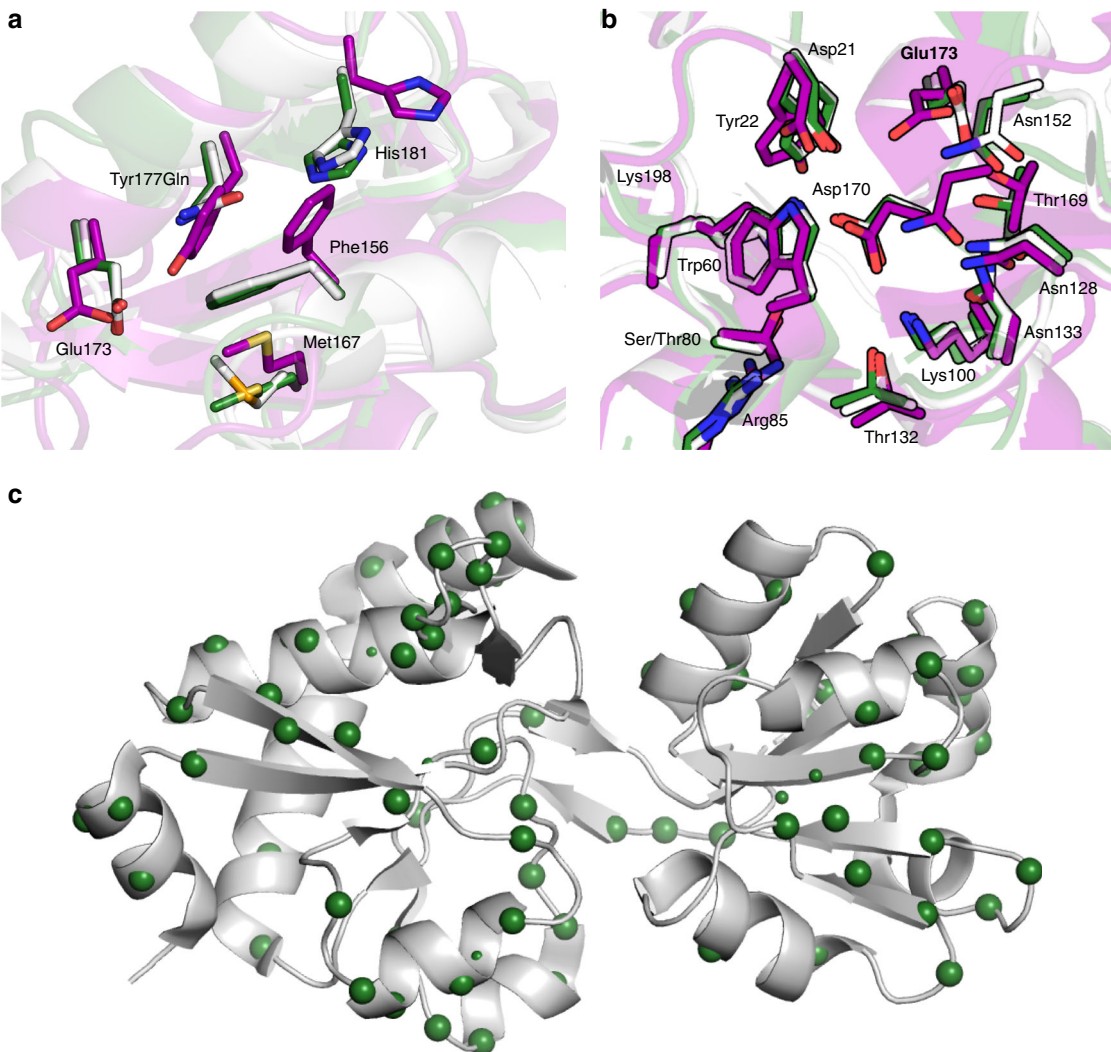

**Fig. 2 Crystal structures of AncCDT-3/P188, AncCDT-5 and *Pa*CDT. a** Structural overlay of the small domains of AncCDT-3/L188 (purple, PDB 5JOS), AncCDT-5 (green) and *Pa*CDT (white, PDB 3KBR). The Tyr177Gln substitution that occurs between AncCDT-3 and AncCDT-5 contributes to the repositioning of Glu173. **b** Structural alignment of the active sites of AncCDT-3/L188 (purple, PDB 5JOS), AncCDT-5 (green) and *Pa*CDT (PDB 3KBR, white). The small domain (residues 99–194) and large domain (residues 1–98 and 195–236) of AncCDT-3/L188 and AncCDT-5 were individually aligned with the corresponding domains of *Pa*CDT. The sidechains of the 15 inner-shell residues identified by Clifton et al.[40] are shown as sticks (Gly131 not shown), highlighting identical active site residues and side-chain conformations between AncCDT-5 and *Pa*CDT. The HEPES molecules present in the crystal structures are omitted for clarity. **c** The locations of the substitutions (green spheres) between AncCDT-5 and *Pa*CDT are shown projected onto the structure of *Pa*CDT (PDB 3KBR).

affecting catalysis by altering the open/closed dynamics of these proteins.

**Solution dynamics of AncCDT-1, evolutionary intermediates and *Pa*CDT.** Next, we investigated whether the accumulation of substitutions between AncCDT-1 and *Pa*CDT at positions remote from the active site could have affected catalytic activity by altering the open/closed dynamics of these proteins. To do so, we used DEER distance measurements and MD simulations to investigate whether the distribution of the open and closed solution states differed between the ancestral amino acid-binding protein (AncCDT-1), the intermediate low-activity CDTs (AncCDT-3, AncCDT-5) and the efficient extant enzyme, *Pa*CDT.

*AncCDT-1.* First, we confirmed that AncCDT-1 was monomeric (>95%) in solution using size-exclusion chromatography multi-angle light scattering (SEC-MALS; Supplementary Fig. 2a).

We then assessed the solution conformations of AncCDT-1 using DEER distance measurements with Gd(III)-tags introduced through unnatural amino acid (UAA) mutagenesis (Supplementary Fig. 3). Amber stop codons were used to incorporate the UAA AzF into the small (Lys138) and large (Gln68) domains of AncCDT-1. These sites were chosen because they (i) are known to be located in regions of the protein that undergo substantial changes in their relative distance upon ligand binding[40], (ii) are located on rigid α-helices, (iii) are surrounded by residues that will minimise unwanted tag motions via packing and (iv) are solvent-exposed, which minimises the risk that mutagenesis and tagging at these positions will significantly affect the protein conformations. The mutant protein heterologously expressed in soluble form in *E. coli*, yielding ~15 mg of purified protein per litre of cell culture. To remove any co-purified ligands, on-column protein denaturation and refolding were performed to obtain ligand-free samples. This is referred to as refolded AncCDT-1. The AzF residues were modified with

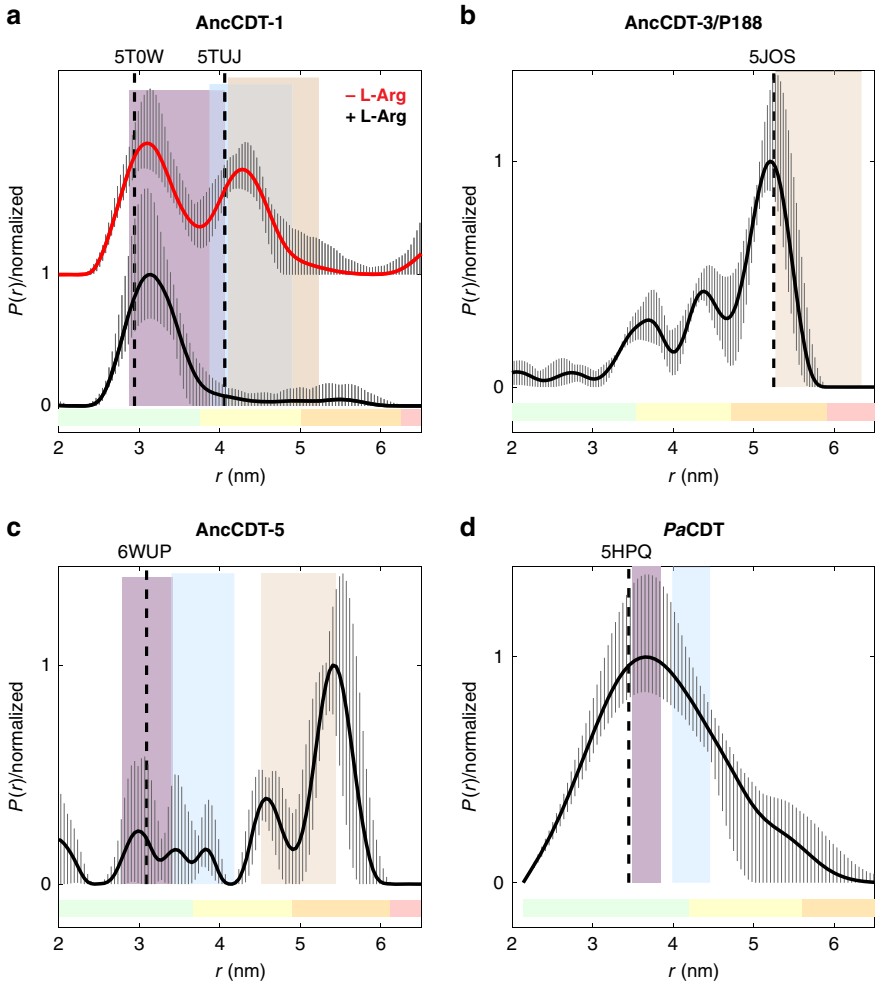

**Fig. 3 Solution dynamics of AncCDT-1, AncCDT-3/P188, AncCDT-5 and *Pa*CDT probed by DEER.** Gd(III)–Gd(III) distance distributions for samples of propargyl-DO3A-Gd(III)-tagged **a** refolded AncCDT-1 (-/+ L-Arg, tagged at positions 68 and 138), **b** natively purified AncCDT-3/P188 (tagged at positions 68 and 138), **c** natively purified AncCDT-5 (tagged at positions 68 and 138) and **d** natively purified *Pa*CDT (tagged at positions 68 and 139 and diluted by a factor of 10 using unlabelled *Pa*CDT). Distance distributions were obtained using DeerAnalysis[77]; the solid lines represent the distributions with the best r.m.s.d. from the experimental data, and the striped regions represent the variation of alternative distributions (± 2 times the standard deviation) obtained by varying the parameters of the background correction and noise. The coloured bars at the bottom of each panel reflect the reliability of each region of the distribution, corresponding to the DEER evolution time used, as defined in DeerAnalysis (pale green: the shape of the distance distribution is reliable; pale yellow: the mean distance and distribution width are reliable; pale orange: the mean distance is reliable; pale red: long-range distance contributions may be detectable, but cannot be quantified.). The original DEER data and DEER form factor traces are shown in Supplementary Figs. 5, 7 and 8. Vertical dashed lines represent the Gd(III)–Gd(III) distances estimated by modelling propargyl-DO3A-Gd(III) tags onto the crystal structures of each protein. The shaded areas represent the range of calculated Gd(III)–Gd(III) distances when propargyl-DO3A-Gd(III) tags are modelled onto a number of individual MD snapshots representing the closed (magenta), open (blue) and wide-open (orange) states.

propargyl-DO3A-Gd(III) tags as previously described (Supplementary Fig. 3a)[41,51], and mass spectrometry analysis indicated that the tag ligation yields were sufficient to deliver at least 75% doubly tagged samples, except for the AncCDT-1 mutant at sites 138 and 161, where at least 25% of the sample was doubly tagged (Supplementary Fig. 4).

DEER measurements of refolded AncCDT-1-propargyl-DO3A-Gd(III) revealed a distance distribution with two dominant peaks with maxima corresponding to Gd(III)–Gd(III) distances of 3.2 nm and 4.4 nm (Fig. 3a, Supplementary Table 2 and Supplementary Fig. 5a, b, i). When saturating concentrations of L-arginine were added to the refolded sample, the distance distribution revealed an almost complete shift to the shorter distance (Fig. 3a), as expected for a ligand-induced open-to-closed transition. Together, these data indicate that the Gd (III)–Gd(III) distance varies by ~1.2 nm between the closed state and any open conformations and that the DEER approach

employed here can be used to differentiate between distinct conformational substates of an SBP. Experiments with natively purified protein rather than unfolded/refolded protein showed similar results (Supplementary Fig. 5a, b, i), except that a greater proportion of the protein adopted the closed state (3.2 nm Gd (III)–Gd(III) distance) in the absence of exogenous L-arginine than in the sample subjected to unfolding and renaturation. This was expected, as the protein is known to co-purify from *E. coli* with some ligand-bound, which stabilises the closed conformation[40]. A ligand-induced shift towards a more closed state was also observed when L-arginine was added to a sample of a lysine-arginine-ornithine binding protein from *Salmonella enterica* (*Se*LAOBP, Supplementary Fig. 6).

We also performed experiments on AncCDT-1 that had been tagged at two sites on either the large domain (positions 68 and 219, Supplementary Fig. 5c) or small domain (positions 138 and 161; Supplementary Fig. 5d). These each showed a peak in the

distance distribution that was in agreement with Gd(III)–Gd(III) distance predictions based on crystal structures of AnCDT-1 (Supplementary Fig. 5g, h). These results confirm that the two distances measured for refolded AncCDT-1 are due to rigid-body motions of the two domains, rather than any intra-domain flexibility or tag dynamics.

We next sought to correlate the distance changes observed through DEER to structural changes in the protein. We previously solved the crystal structure of AncCDT-1 bound to L-arginine (PDB 5T0W), which has a Cα–Cα distance between Gln68 and Lys138 of 2.6 nm (equivalent to a Gd(III)–Gd(III) distance of ~2.9 nm, Supplementary Fig. 5e). We also solved a structure of the protein in an open state (PDB 5TUJ), in which the Cα–Cα distance between Gln68 and Lys138 is 3.4 nm (Gd(III)–Gd(III) distance of ~4.1 nm, Supplementary Fig. 5f). This crystal structure presents a snapshot of an open conformation, but additional open states may be accessible to the protein in solution. To probe these, we also explored the solution dynamics of AncCDT-1 using MD simulations initiated from the closed X-ray crystal structure of AncCDT-1 (PDB 5T0W) with the ligand, L-arginine, removed. These were conducted solely to identify conformational substates that can be accessed on the ns–μs timescale (i.e., easily accessible, low-energy states), not to obtain quantitative predictions of the solution dynamics or the relative occupancy of each state. The simulations suggest that AncCDT-1 can easily access three dominant conformational substates (Fig. 4a and Supplementary Table 2): the closed state corresponding to PDB 5T0W (68–138 Cα–Cα distance range of 2.5–2.9 nm; equivalent to a Gd(III)–Gd(III) distance range of ~2.9–3.4 nm), an intermediate open state corresponding to PDB 5TUJ (68–138 Cα–Cα distance 3.2–3.7 nm; equivalent to a Gd(III)–Gd(III) distance range of ~3.9–4.9 nm) and an additional wide-open state (68–138 Cα–Cα distance 3.6–4.2 nm; equivalent to a Gd(III) distance range of ~4.1–5.2 nm).

The results obtained by DEER suggest that the dominant states are indeed the closed state and an open conformation closely related to the open state observed through protein crystallography, but which might also include the wide-open state, as there is significant overlap between the modelled Gd(III)–Gd(III) distances in the snapshots obtained from MD. Although we do not have crystallographic evidence that AncCDT-1 samples the wide-open state, analogous wide-open states have been observed in crystal structures of extant L-arginine-binding proteins, such as that from *Thermotoga maritima* (*Tm*ArgBP, PDB 4PRS)[38], which has a Cα–Cα distance of 4.0 nm between positions equivalent to 68–138 in AncCDT-1 (which is similar to the 3.6–4.2 nm range observed for the wide-open state in AncCDT-1 MD simulations). Indeed, it has been suggested that the specific open states of SBPs observed crystallographically can be dictated by crystal packing interactions[16–19]. The wide-open state thus is a part of the conformational landscape for L-arginine-binding proteins, although we could not unambiguously detect it in our DEER experiments on AncCDT-1 owing to overlap with the open state. Altogether, the data from DEER, protein crystallography and MD demonstrate that AncCDT-1 can sample both open and closed states in the absence of ligands, and that ligand binding shifts the conformational equilibrium towards the closed state.

*AncCDT-3 and AncCDT-5.* Considering the significant difference in dehydratase activity between AncCDT-3, AncCDT-5 and *Pa*CDT, as well as our inability to fully rationalise these differences through protein crystallography, we next used DEER and MD to investigate the solution conformational distributions of these proteins. Like AncCDT-1, SEC-MALS indicated that AncCDT-3/P188 and AncCDT-5 were predominantly monomeric in solution (>95%), as reported previously[40]

(Supplementary Fig. 2b, c). We tagged the small and large domains of AncCDT-3/P188 and AncCDT-5 at sites 68 (large domain) and 138 (small domain) in the same way as AncCDT-1 and performed DEER experiments (Fig. 3b, c and Supplementary Fig. 7a, b). For natively purified AncCDT-3/P188, the DEER distance distributions showed a maximum that corresponded to a Gd(III)–Gd(III) distance of ~5.2 nm (Fig. 3b and Supplementary Fig. 7a). Likewise, AncCDT-5 displayed the most prominent distance-distribution peak at ~5.4 nm (Fig. 3c and Supplementary Fig. 7b). Similar results were obtained for the refolded proteins (Supplementary Fig. 7a, b), where the observed shifts are within experimental error. Although the confidence ranges identify clear peak maxima, we also calculated mean distance distributions (caption of Supplementary Fig. 7) to further assess the reliability of these peak positions due to the limited evolution time with respect to the long distance observed. This resulted in shifts of at most about 0.25 nm.

We previously solved a crystal structure of AncCDT-3/L188 and found a conformation analogous to the wide-open conformation observed in the MD simulation of AncCDT-1 (PDB 5JOS)[40]. In this structure, the Q68-R138 Cα–Cα distance is 4.2 nm (Gd(III)–Gd(III) distance: ~5.3 nm). In triplicate 500 ns MD simulations initiated from the open AncCDT-3/L188 and AncCDT-3/P188 structures (after modelling the L188P mutation), the protein did not sample the closed state at all across the combined 1.5 μs of simulation time and the wide-open state was the major conformation (Fig. 4b), exhibiting a Cα–Cα distance range of 4.0–4.8 nm (Gd(III)–Gd(III) distance range: 5.0–6.3 nm).

Unlike the wide-open crystal structure of AncCDT-3/L188, the 1.49 Å crystal structure of AncCDT-5 determined in this work adopts the closed conformation (a HEPES molecule is bound in the active site), with a Q68-R138 Cα–Cα distance of 2.8 nm (Gd(III)–Gd(III) distance: 3.1 nm). Beginning from this closed state, we again performed triplicate 500 ns MD simulations after removing the HEPES molecule. The simulations predict that the protein rapidly adopts a wide-open conformation, with a Q68-R138 Cα–Cα distance range of 3.6–4.2 nm (Gd(III)–Gd(III) distance range: 4.5–5.4 nm) that is analogous to the wide-open conformation seen in the MD simulations of AncCDT-1, the crystal structure of the L-arginine-binding protein from *T. maritima*, and the crystal structure of AncCDT-3 (Fig. 4c). An intermediate state, analogous to the open crystal structure of unliganded AncCDT-1 was also briefly sampled, which corresponded to a Q68-R138 Cα–Cα distance range of 3.0–3.4 nm (Gd(III)–Gd(III) distance range: 3.4–4.2 nm). The combination of the DEER, crystallographic and MD data for the experiments performed on AncCDT-3 and AncCDT-5 suggest that both proteins predominantly access a wide-open state in solution. Smaller peaks in the distance distribution could correspond to minor fractions of intermediate states but are too small to be unambiguous.

*Pa*CDT. We next aimed to probe the open/closed dynamics of *Pa*CDT. SEC-MALS experiments confirmed that *Pa*CDT is primarily homotrimeric in solution[40], with a small population (<5%) of higher-order or aggregated species (Supplementary Fig. 2d), which complicates the measurement of intra-monomer distances by DEER experiments. To confirm the existence of the tagged trimer, we first measured samples with propargyl-DO3A-Gd(III) tags at single AzF residues of each *Pa*CDT monomer, at either the large domain (position 68) or the small domain (position 139). The maxima of the DEER distance distributions measured were consistent with those expected for the trimeric structure observed by X-ray crystallography (3.5 nm vs. 3.3 nm for proteins tagged at position 68, 5.4 nm vs. 4.6 nm for samples tagged at position 139; Supplementary Fig. 8a, b). This indicated that the trimeric

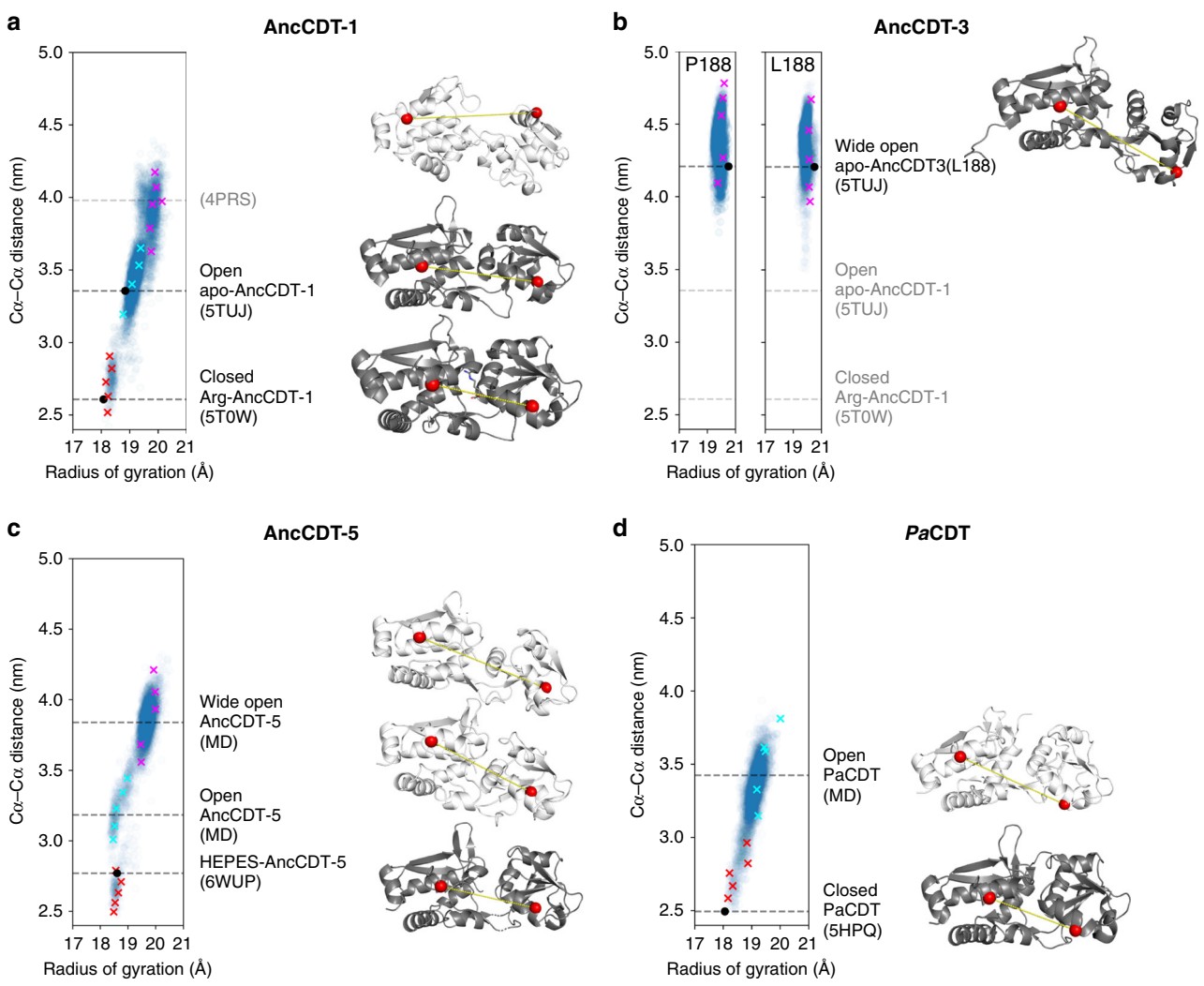

**Fig. 4 Conformational substates captured by MD and crystallography.** Conformational substates of (**a**) AncCDT-1, (**b**) AncCDT-3, (**c**) AncCDT-5 and (**d**) *Pa*CDT captured by MD and X-ray crystallography. The plot of Cα–Cα distances between tagged residues vs. radius of gyration obtained from MD simulation snapshots highlights populations (or lack thereof) of closed, open and wide-open conformational states. Each data point represents a single frame of the MD simulation (sampled every 0.1 ns). Crosses represent snapshots of closed (red), open (cyan) and wide-open (magenta) states that were used to model Gd(III)–Gd(III) distances. Representative structures of each of these states are shown on the right and highlight the extent of the open/closed transition, with distances between Cαs (red spheres) of tagged sites shown by yellow lines. Dark grey structures are crystal structures (corresponding to black dots on the plot), while white structures are MD snapshots. The time evolution of Cα–Cα distances are shown in Supplementary Fig. 10.

structure was preserved under the conditions of the DEER experiments, including sample preparation and freezing. When we attempted to measure the open/closed distribution via labelling at sites on both the large and small domain (68 and 139, respectively) the observed Gd(III)–Gd(III) distance distribution was very broad as was expected due to multiple inter-chain Gd(III)–Gd(III) distances in a trimer (Supplementary Fig. 8c). In order to extract the inter-domain distance from a single *Pa*CDT monomer, a protein sample labelled at sites 68 and 139 was diluted tenfold with unlabelled protein and allowed to equilibrate at room temperature, with the aim of obtaining a mixture in which there would be a ~24%, 3% and 0.1% chance of encountering a trimer consisting of 1, 2 or 3 tagged monomers, respectively. DEER experiments on samples prepared in this way yielded a significantly narrower peak that indicated 3.5 nm was the predominant Gd(III)–Gd(III) distance (Fig. 3d and Supplementary Fig. 8d).

We again used protein crystallography and MD simulation to identify stable conformations of *Pa*CDT. There are three crystal

structures, one with HEPES bound at the active site (Structural Genomics Project, PDB 3KBR) and two with acetate bound at the active site[40]. The highest resolution acetate-bound structure (PDB 5HPQ) and the HEPES-bound structure (PDB 3KBR) display Q68-A139 Cα–Cα distances of 2.5 nm (Gd(III)–Gd(III) distance: ~3.5 nm). A MD simulation initiated from the trimeric X-ray crystal structure of acetate-*Pa*CDT (PDB 5HPQ), with acetate removed, indicated that the monomers can sample both closed (Cα–Cα distance range of 2.5–3.0 nm; equivalent to a Gd(III)–Gd(III) distance range of ~3.5–3.9 nm) and open (Cα–Cα distance range of 3.2–3.8 nm; equivalent to a Gd(III)–Gd(III) distance range of ~4.0–4.5 nm) conformations, but there was no evidence of a wide-open conformation (Fig. 4d). The Gd(III)–Gd(III) distances predicted for the closed and open states notably fall either side of the broad peak centred at 3.9 nm. This could be consistent with a broad distribution of rapidly interconverting open and closed states that have been snap-frozen, in contrast with the AncCDT-1 DEER analysis, which displayed two distinct peaks corresponding to similar distances.

All three crystal structures of *Pa*CDT display closed conformations, which leave limited access to the active site. To allow the exchange of substrate and product, the enzyme must also populate open conformations, which are likely related to those obtained in the MD simulations. Interestingly, however, the wide-open conformation, which appears in MD simulations of AncCDT-1, AncCDT-3 and AncCDT-5, was not significantly sampled. This is again consistent with the DEER measurements, where the prominent peaks at ~5.4 nm in AncCDT-3 and AncCDT-5 were absent for *Pa*CDT. Altogether, the combined use of DEER, crystallography and MD simulations suggests that the wide-open conformation observed in the ancestral reconstructions of CDT (computationally observed in all variants and empirically observed in AncCDT-3, AncCDT-5 and orthologs of AncCDT-1) has been largely eliminated from the conformational landscape of the extant *Pa*CDT. This provides a molecular explanation for the observed change in $K_M$ between AncCDT-5 and *Pa*CDT; despite their identical substrate-binding sites when closed, the open ligand-binding state of *Pa*CDT likely has a higher affinity for the substrate as it is much closer to the conformation of the Michaelis complex than the wide-open state that is predominantly populated in AncCDT-3 and AncCDT-5.

**Structural basis for the increased dehydratase activity of *Pa*CDT.** Having established that the conformational sampling of *Pa*CDT is different from AncCDT-5, we compared the structures of AncCDT-5 and *Pa*CDT to ascertain the molecular basis for this change. Structural comparison between the two proteins revealed three regions with substantial differences (unlike the substrate-binding site) that could collectively stabilise the closed state and/or destabilise the wide-open state. First, in comparison to AncCDT-5, which terminates at Lys235 (although it is not present in clear electron density), *Pa*CDT has an additional nine amino acids at the C-terminus (RWPTAHGKL), of which the first four and six residues produce clear electron density in crystal structures of *Pa*CDT (PDB 5HPQ and 6BQE, respectively). As shown in Fig. 5a, these amino acids interact extensively with both the small and large domains, substantially increasing the number of van der Waals and hydrogen bond interactions between the two domains in the closed and open, but not wide-open, conformations.

Second, one of the most substantial structural differences between AncCDT-5 and *Pa*CDT is the oligomerization of *Pa*CDT, which forms a trimer (Fig. 5b). A number of key residues that contribute to the inter-subunit interfaces in the crystal structures of *Pa*CDT are not found in AncCDT-5, explaining this difference in oligomeric states. For example, the substitutions T97R, M221Q and L217H introduce specific electrostatic interactions between chains in *Pa*CDT, while H195F, H218I and P83L appear to improve packing between the subunits. However, oligomerization is a complex process and remote and unpredictable mutations can also have a substantial influence on oligomeric states[52]. Accordingly, we cannot exclude the possibility that additional mutations remote from the interface also play a role.

In the resulting trimeric form of *Pa*CDT, the small domain remains free to move, allowing fluctuation between the closed and open states. However, there are inter-subunit contacts at the hinge region that appear to preclude some of the conformational changes expected in the wide-open state. Specifically, the crystal structure of the wide-open conformation of AncCDT-3 shows that the hinge region with His195 at the centre is fully solvent-exposed and extended. The equivalent region of *Pa*CDT, however, is in a retracted conformation, with Phe195 (chain A; the equivalent position to His195 in AncCDT-3) being stabilised

in a buried hydrophobic pocket formed by Ile218 (chain A) and the non-polar region of Ser222 (chain A), as well as the non-polar regions of two hydrogen-bonded pairs, Arg97 (chain A):Gln221 (chain B) and Glu197 (chain A):His217 (chain B). The positioning of these residues and interactions with the neighbouring subunit would likely stabilise *Pa*CDT in the closed and open conformations, while limiting the sampling of wide-open states.

Finally, for the proteins to adopt the wide-open state, the region between Val187 and Pro191 must undergo a conformational change, from the tightly kinked conformation seen in the closed structures of AncCDT-5 and *Pa*CDT (PDB 5HPQ, 6BQE; it is in an altered conformation due to crystal packing interactions in 3KBR), to an extended conformation seen in the crystal structure of the wide-open conformation of AncCDT-3 (Fig. 5c). In AncCDT-3/5, the region between Val187 and Pro191 is relatively flexible, consisting of polar and relatively small hydrophobic residues (Leu/Val/Glu in AncCDT-3; His/Val/Glu in AncCDT-5). In contrast, in *Pa*CDT, the central residue in the kink is mutated to proline, and the neighbouring glutamine forms a hydrogen bond to Arg186, stabilising the kinked structure. Further support for the hypothesis that this region is important in conformational sampling and dehydratase activity can be found in the observation that (i) the P188L mutation in AncCDT-3 causes a large decrease in $K_M$[40], (ii) it was observed to be a mutational hotspot during the directed evolution of ancestral CDTs for increased dehydratase activity[40] and (iii) this region has been recognised to be important for dynamics and affinity in other SBPs such as MBP[21,30].

The combination of these three structural effects is likely sufficient to substantially change the relative conformational free energy of the closed, open and wide-open states, consistent with the DEER and MD results. Interestingly, none of the structural changes observed between AncCDT-3 and *Pa*CDT preclude *Pa*CDT from adopting the open conformation, but essentially prevent it from adopting the wide-open conformation. Given the large number of changes between AncCDT-5 and *Pa*CDT, it is impossible to ascribe the difference in populations of open and wide-open conformations to a single mutation or structural rearrangement. Indeed, recent work has highlighted the gradual nature by which protein conformational sampling can change, and the overlapping and often redundant effects of mutations[49].

## Discussion

SBPs have become a model system for understanding the role of rigid-body motions in determining ligand and substrate-binding affinity, specificity and kinetics. Early crystallographic studies suggested that SBPs act as simple binary switches, adopting a closed conformation when in complex with ligands and an open conformation in the ligand-free state[4]. More recently, however, NMR, FRET and MD studies have probed the intrinsic dynamics of SBPs, showing that they can adopt closed and semi-closed states in the absence of ligands[20–29], and have provided evidence that the extent of the open-closed dynamics evolves to support specific binding properties and function. Indeed, altering the open-closed dynamics of SBPs is now also being used as an additional parameter for engineering new binding functions into SBPs[21] and the development of novel SBP-based biosensors[53]. Such efforts, however, are complicated by the fact that very subtle changes to protein motion can alter SBP function and the structural basis for changes in these open-closed dynamics are poorly understood. Here, we have expanded on this work, demonstrating that SBPs of this protein fold can access three relatively well-defined conformational substates, including a wide-open state for which the possible physiological role must still be determined. In view of the ease with which it is sampled in

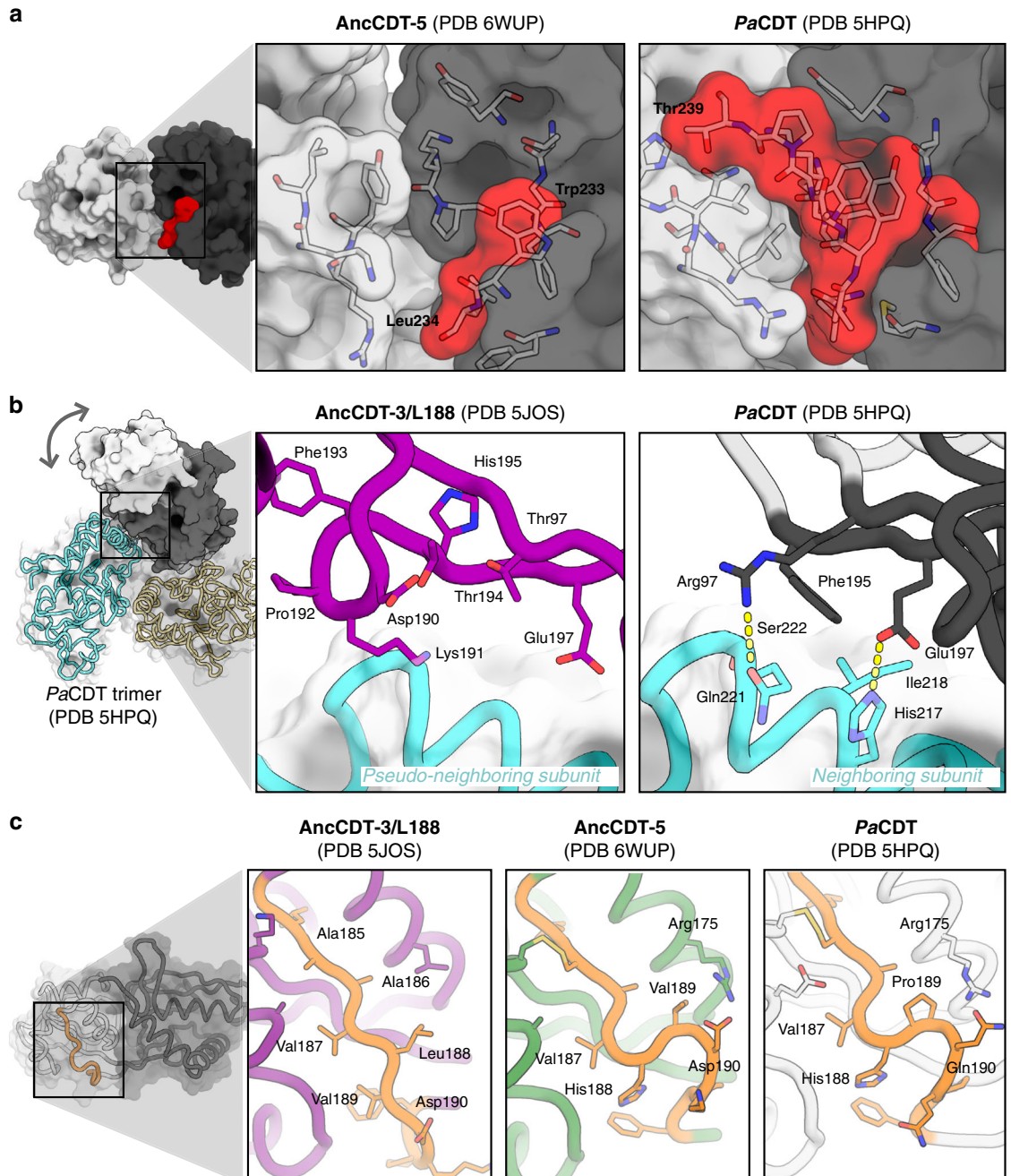

**Fig. 5 Structural basis for the change in conformational sampling. a** Comparison of the crystal structures of AncCDT-5 (left) and *Pa*CDT (5HPQ, right) highlight how the C-terminal extension (residues beyond 232 are highlighted in red) in *Pa*CDT facilitates additional interactions between the small (white) and large (grey) domains. **b** *Pa*CDT is a trimer in solution, and analysis of the crystal structures of *Pa*CDT (e.g., 5HPQ, shown here) reveals several interactions between the hinge region of chain A with the large domain of chain B that are likely to prevent the wide-open conformation from being sampled. These include packing of Phe195 against the neighbouring subunit, and polar interactions between neighbouring residues. Overlay of the crystal structure of AncCDT-3/L188 (purple, PDB 5JOS) further highlights how the wide-open state is likely to be incompatible with the trimeric arrangement of *Pa*CDT. **c** The flexibility of the hinge region is known to control the open/closed dynamics of SBPs. Sequence and structural comparison of these regions in AncCDT-3/L188 (purple, PDB 5JOS), AncCDT-5 (green, PDB 6WUP) and *Pa*CDT (white, PDB 5HPQ) highlight that this is an intrinsically flexible region, which is likely affected by mutations.

MD simulations and unambiguously observed by DEER experimentally in two of the native proteins we studied (AncCDT-3 and AncCDT-5), as well as the apo-state of the L-arginine-binding protein from *T. maritima*[38], it is unlikely to be an artefact. The apparently facile sampling of both closed and wide-open states could explain why FRET experiments, which have been conducted with fluorescent tags attached to either domain, sometimes report smaller changes in signal upon ligand binding than would be expected if a single open conformation was fully populated in the apo-protein[54,55].

We previously suggested that mutations accumulated during evolution most likely increased the dehydratase activity of *Pa*CDT by shifting the conformational equilibrium away from the catalytically unproductive open state to the catalytically competent

closed state[40]. Crystal structures and MD simulations provided some preliminary evidence that this is the case, but now we have strong quantitative evidence in solution. First, the crystal structure of AncCDT-5 revealed identical active/binding sites of AncCDT-5 and *Pa*CDT, demonstrating that amino acid substitutions remote from the active site can have very large effects on activity via mechanisms other than controlling the structure of the active site. Second, the dramatic shift in the open/closed equilibrium of the proteins is the main change along the evolutionary trajectory. These results show that the wide-open state, which is the lowest-energy state in AncCDT-3 and AncCDT-5 and is unlikely to play any catalytic role owing to low affinity for substrate owing to the spatial separation between the binding site/catalytic residues, appears to be depopulated in the final evolutionary step between AncCDT-5 and *Pa*CDT. Third, the extant and efficient *Pa*CDT appears to exist in a broad distribution between a closed state optimised for catalysis of the chemical step in the reaction and a moderately open state that retains the shape of the binding pocket but opens a pathway for substrate/product diffusion. The structural analysis provides a molecular explanation for this shift in conformational sampling, with a number of structural changes stabilising the open/closed states at the expense of the wide-open state. Interestingly, this seems to have occurred at least in part through oligomerization, suggesting that evolution can favour oligomerization both for stability and activity[52].

The role of the wide-open state along the evolutionary trajectory from AncCDT-1 to *Pa*CDT is more difficult to interpret. We observe this state in crystal structures of extant relatives of AncCDT-1, it is stable for several hundred ns in MD simulations of AncCDT-1 and (because of the overlap of the DEER signals between the wide-open and open states) it is also consistent with the DEER data. However, it is clear that there was a shift towards the wide-open state after AncCDT-1 (as seen in AncCDT-3 and AncCDT-5). In our previous work[40], we were unable to identify a natural ligand for the non-enzymatic binding protein AncCDT-2 (which predates AncCDT-3 and AncCDT-5); although it had lost its ancestral arginine-binding function, it (and its descendants such as Pu1068 from the aquatic bacterium *Pelagibacter ubique*) retain the genomic organisation and biophysical properties of solute-binding proteins. Thus, we speculate that the shift towards the wide-open state in the intermediate ancestors could have resulted from the selective pressure for the (still unknown) solute-binding function that preceded the evolution of CDT activity.

The findings from this work add to growing evidence that the modification of protein dynamics often plays an important role in the evolution of new protein function. While the current work provides a unique example for pronounced, historical conformational changes between a non-catalytic ancestral protein and an efficient enzyme, our findings mirror those of other work that highlight how catalytic activity can be enhanced along evolutionary trajectories through the accumulation of remote substitutions that alter conformational dynamics[49,56–60]. Modified protein dynamics are not mutually exclusive with the well-established concept of evolution of protein function via specific changes in the shape or electrostatic character of active/binding sites, but serve as a complementary mechanism that may often overlap with these other more easily observable changes.

In addition, this work expands the scope of the DEER approach[15,61–65] for use in this context by using Gd(III) spin labels; it is the first time that the propargyl-DO3A-Gd(III)/DEER approach has been used to study rigid-body protein dynamics and it demonstrates the excellent performance of the propargyl-DO3A-Gd(III) tag. The tag is independent of cysteine residues as it can be ligated to AzF residues and the small size, lack of net charge and hydrophilic character of the DO3A-Gd(III) complex

disfavours specific interactions with the protein surface, thus increasing the reliability of computational predictions of the Gd(III)–Gd(III) distance distributions obtained with this tag[41]. This study highlights its efficiency (the DEER measurements requiring only nanogram amounts of protein) in studying the protein dynamics of a series of states along an evolutionary trajectory and how it afforded distribution widths that were sufficiently narrow to observe the co-existence of open and closed protein conformations that differ by only about 1 nm, to show the presence of two conformations simultaneously sampled in solution and to detect ligand-induced conformational changes.

Considering that DEER data are recorded of frozen solutions, it cannot be ruled out that conformational equilibria established at room temperature change during the snap-freezing of the samples. In principle, the freezing point of 20% deuterated glycerol solutions is at most 10° below 0 °C[66], but the actual temperature at which the dynamics stopped is difficult to determine. As all samples were frozen in the same way, however, the conformational differences observed between the samples refer to similar conditions. It is thus significant that the energy landscape of the extant enzyme *Pa*CDT led to the freezing out of a continuous range of conformations, whereas that of the amino acid-binding protein AncCDT-1 funnelled the protein into two clearly distinct conformations. Importantly, the distances measured by DEER here are in good agreement with distances predicted from the crystal structures and states obtained from MD simulations.

In this work, we have studied the structural dynamics of a series of states along an evolutionary trajectory from an ancestral L-arginine solute-binding protein (AncCDT-1) to an extant dehydratase (*Pa*CDT). The challenge of accurately defining the conformational landscape of a protein is substantial: virtually all approaches have serious limitations or are so demanding that they are not practical for studying more than a single protein. Here, by following an integrated structural biology approach involving the combined use of the three methods, the shortcomings of each approach (crystal packing forces in single crystals, imperfect force fields in MD simulations, potential overlap in the signals from different states for DEER measurements, etc.) could be compensated for. We compared crystal structures of AncCDT-3, a new structure of AncCDT-5, and *Pa*CDT, which revealed that second-shell residues alter the active site configuration between AncCDT-3 and AncCDT-5, but that AncCDT-5 and *Pa*CDT display large differences in catalytic activity despite identical active site configurations. Using a propargyl-DO3A-Gd(III) tag and W-band DEER distance measurements, we were able to experimentally assess the distribution of different conformational substates in frozen solutions along this evolutionary trajectory, with reference to structures obtained through protein crystallography and MD simulations. These data provide new insight into how proteins can evolve new functions: while the correct active site configuration must be established, this study shows that the ability of an enzyme to adopt a range of conformational states that are tailored to its catalytic role is equally important. This result contributes to the growing body of literature that suggests molecular evolution is a relatively smooth process, whereby conformational substates that are not productive are gradually depopulated while beneficial substates are enriched.

## Methods

**Protein numbering convention.** Residues of all proteins are referred to using the equivalent position in the reconstructed ancestral sequences (i.e., not including N-terminal tags), as described in the previous work[40] (see sequence alignment, Supplementary Fig. 9). As a consequence, residue numbers mentioned in this work differ from the residue numbers in the published PDB files of 3KBR (shift of −25), 5HPQ (−11), 6BQE (−11), 5T0W (−11), 5TUJ (−11), 5JOS (−11) and 6WUP (−11).

**Materials**. The genes encoding *Se*LAOBP (UniProt: P09551), AncCDT-1, AncCDT-3/P188, AncCDT-3/L188, AncCDT-5 and *Pa*CDT (UniProt: Q01269, residues 26–268) in pDOTS7 vectors (a derivative of pQE-28L, Qiagen) were obtained from previous work[32,40] (DNA sequences supplied in Source Data file). The Δ*pheA* strain of *E. coli* K-12 from the Keio collection (strain JW2580-1) was obtained from the Coli Genetic Stock Center (Yale University, CT).

**Prephenate dehydratase assay**. Protein expression and purification, prephenate preparation and spectrophotometric assays were performed as described previously[40].

To prepare sodium prephenate, 40 mM barium chorismate (Sigma, 60–80% purity) was mixed with an equimolar amount of 1 M $Na_2SO_4$. An equal volume of 100 mM $Na_2HPO_4$ (pH 8.0) was added to the mixture, the $BaSO_4$ precipitate was removed, and the resulting sodium chorismate solution was heated at 70 °C for 1 h to yield sodium prephenate. The concentration of prephenate was measured by quantitative conversion of prephenate to phenylpyruvate under acidic conditions (0.5 M HCl, 15 min, 25 °C) and spectrophotometric determination of phenylpyruvate concentration by measuring absorbance at 320 nm, as described previously[61].

Proteins were expressed in Δ*pheA* cells (from pDOTS7 plasmids) to exclude the possibility of contamination with endogenous prephenate dehydratase. Cells were grown in Terrific Broth (TB) media at 37 °C to OD$_{600}$ 0.8, induced with 0.5 mM IPTG and incubated for an additional 12 h at 30 °C. Cells were resuspended in equilibration buffer (50 mM $NaH_2PO_4$, 500 mM NaCl, 20 mM imidazole, pH 7.4), lysed by sonication and fractionated by ultracentrifugation (24,200 × *g* for 1 h at 4 °C). The supernatant was filtered through a 0.45-μm filter and loaded onto a 5 mL HiStrap HP column (GE Healthcare, USA) equilibrated with equilibration buffer. The column was washed with 50 mL of equilibration buffer and 25 mL of wash buffer (50 mM $NaH_2PO_4$, 500 mM NaCl, 44 mM imidazole, pH 7.4), and the target protein was eluted in 25 mL of elution buffer (50 mM $Na_2HPO_4$, 500 mM NaCl, 500 mM imidazole, pH 7.4). Proteins were concentrated using a centrifuge filter (Amicon Ultra-15 filter unit with 10 kDa cut-off) and purified by size-exclusion chromatography (SEC) on a HiLoad Superdex 75 16/600 column (GE Healthcare, USA), eluting in SEC buffer (20 mM $Na_2HPO_4$, 150 mM NaCl, pH 7.4). Protein purity was confirmed by SDS-PAGE, and protein concentrations were measured spectrophotometrically using predicted molar absorption coefficients[62].

Prephenate dehydratase activity of each protein was determined by end-point spectrophotometric measurement of phenylpyruvate formation, as described previously[61]. Protein solutions were prepared in 20 mM $Na_2HPO_4$, 150 mM NaCl (pH 7.4), and prephenate solutions were prepared in 50 mM $Na_2HPO_4$ (pH 8.0). After equilibration at room temperature (20–25 °C) for 5 min, the reaction was initiated by mixing equal volumes of protein and substrate solutions. Aliquots (50 μL or 100 μL) were regularly removed from the reaction mixture and quenched by the addition of an equal volume of 2 M NaOH. Absorbance at 320 nm was measured using an Epoch Microplate Spectrophotometer (BioTek), and phenylpyruvate concentrations were determined assuming a molar extinction coefficient of 17,500 $M^{-1}$ $cm^{-1}$. Reaction times and enzyme concentrations were adjusted to ensure that <20% conversion of prephenate to phenylpyruvate. The rate of non-enzymatic turnover was subtracted from the observed rate of enzyme-catalysed turnover. GraphPad Prism (8.3.1) was used for fitting a Michaelis–Menten model to the data.

**Measurement of oligomeric state**. IMAC-purified, His$_6$-tagged AncCDT-1, AncCDT-3/P188, AncCDT-5 and *Pa*CDT (expressed in Δ*pheA* cells) were transferred into size-exclusion chromatography multi-angle light scattering (SEC-MALS) buffer (20 mM $Na_2HPO_4$, 150 mM NaCl, pH 7.4) and concentrated using an Amicon centrifuge filter (10 kDa molecular weight cut-off, MWCO). Samples (100 μL) of each protein were loaded at 10 mg mL$^{-1}$ onto a pre-equilibrated Superdex 200 10/300 GL size-exclusion column (GE Healthcare) attached to multi-angle light scattering (DAWN HELEOS 8; Wyatt Technologies) and refractive index detection (Optilab rEX; Wyatt Technologies) units. A flow rate of 0.5 mL min$^{-1}$ was used. The multi-angle detectors were normalised using monomeric bovine serum albumin (Sigma, A1900). A *dn/dc* value of 0.186 g$^{-1}$ was used for each sample. The data were processed using ASTRA 5..3.4 (Wyatt Technologies). Data were collected from a single experiment (*n* = 1).

**Protein sample preparation for DEER experiments**. For DEER experiments, genes encoding each protein were cloned into pETMCSIII[67] and expressed with an N-terminal His$_6$-tag followed by a TEV cleavage site. In order to minimise tag side-chain dynamics, the crystal structures of the native proteins were inspected in PyMOL (The PyMOL Molecular Graphics System, Version 2.0 Schrödinger, LLC.) to select mutation sites where the sidechains of AzF residues were predicted to populate single χ$_1$-angle conformations. Amber stop codons were introduced at these positions (Supplementary Table 3) by a modified QuikChange protocol using mutant T4 DNA polymerase[68]. Primer sequences are provided in Supplementary Table 4.

All proteins for DEER experiments were expressed in RF1-free *E. coli* strain B-95.ΔA cells[69] co-transformed with a plasmid for the aminoacyl-tRNA synthetase/tRNA pair[70] for the incorporation of AzF (Chem-Impex, USA). To minimise the

amount of AzF required (provided at 1 mM), 1 L of cell culture grown in LB medium was concentrated to 300 mL (by centrifugation and resuspension) before induction with 1 mM IPTG. Expression was conducted at 37 °C and limited to 3 h after IPTG induction to minimise the chemical reduction of AzF. Cells were harvested by centrifugation at 5000 × *g* for 15 min, and lysed by passing two times through a French Press (SLM Aminco, USA) at 830 bars. The lysate was centrifuged at 13,000 × *g* for 30 min, and the filtered supernatant was loaded onto a 5-mL Ni-NTA HisTrap HP column (GE Healthcare, USA) equilibrated with binding buffer (50 mM Tris-HCl, pH 7.5, 150 mM NaCl, 5% w/v glycerol). The protein was eluted using elution buffer (50 mM Tris-HCl, pH 7.5, 150 mM NaCl, 5% w/v glycerol, 300 mM imidazole) and fractions were analysed by 12% SDS-PAGE.

To liberate and remove ligand molecules that may have bound during protein expression, the preparation of the second set of protein samples included a denaturation and on-column refolding step. After cell lysis, guanidinium hydrochloride was added to the supernatant to a final concentration of 6 M. The filtered solution was then loaded onto a Ni-NTA column and washed with binding buffer containing 8 M urea to remove any bound ligand. Refolding was achieved using a gradient of binding buffer with decreasing amounts of urea overnight at a flow rate of 0.5 mL min$^{-1}$. Refolded protein was eluted as described above.

His$_6$-tags were removed by digestion with His$_6$-tagged TEV protease[71] in 1:100 molar ratio overnight at 4 °C in TEV cleavage buffer (50 mM Tris-HCl, pH 8.0, 300 mM NaCl and 1 mM β-mercaptoethanol). Finally, the cleaved His$_6$-tag and TEV protease were removed by passing through the Ni-NTA column (pre-equilibrated with binding buffer) prior to ligation with the propargyl-DO3A tag loaded with Gd(III).

Click reactions with the propargyl-DO3A-Gd(III) tag were performed overnight at room temperature as described previously (Supplementary Fig. 3a)[51] and samples were exchanged into EPR buffer (20 mM Tris-HCl, pH 7.5 in $D_2O$) using a Amicon Ultra-15 filter unit (10 kDa MWCO). The tagging yields were assessed by mass spectrometry, using an Elite Hybrid Ion Trap-Orbitrap mass spectrometer coupled with an UltiMate S4 3000 UHPLC (Thermo Scientific, USA).

DEER measurements were done on samples containing 100 μM protein and 20% w/v glycerol-d$_8$ in $D_2O$. A 1.5-fold molar excess of L-arginine was added to one sample of refolded AncCDT-1 and one sample of *Se*LAOBP prior to the DEER measurement. Samples were flash-frozen by quick insertion of the sample holder into the cold (10 K) magnet bore.

**DEER measurements**. DEER measurements were conducted at 10 K on a home-built pulse EPR spectrometer operating at W-band (94.9 GHz)[72,73]. For the doubly labelled AncCDT-1 samples, a variant of the standard four-pulse DEER sequence $(\pi/2(\nu_{obs}) - \tau_1 - \pi(\nu_{obs}) - (\tau_1 + t) - \pi(\Delta\nu_{pump}) - (\tau_2 - t) - \pi(\nu_{obs}) - \tau_2 - echo)$ was used[74,75]. The DEER echo was observed at 94.9 GHz with π/2 and π pulses of 15 ns and 30 ns, respectively, and the field was positioned at the peak of the Gd(III) spectrum. The pump pulse was replaced by two consecutive chirp pulses produced by an arbitrary waveform generator[74–76] which were positioned on both sides of the centre of the Gd(III) spectrum with frequency ranges of 94.5–94.8 GHz and 95–95.3 GHz, respectively. The length of each chirp pulse was 96 ns. The cavity was tuned at 94.9 GHz with a maximal microwave amplitude, ω$_1$/2π, of ~30 MHz. Other parameters used were τ$_1$ = 350 ns, a repetition rate of 0.8 ms, and τ$_2$ typically ranging between 4 and 7 μs for samples with shorter and longer distances, respectively. The delay *t* was varied during the experiments. For the rest of the samples, a reverse DEER (rDEER) sequence $(\pi/2(\nu_{obs}) - \tau_1 - \pi(\nu_{obs}) - (\tau_1 - t) - \pi(\nu_{pump}) - (\tau_2 + t) - \pi(\nu_{obs}) - \tau_2 - echo)$ was used to reduce artefacts[74]. The chirp pump pulses and the positioning of the observed and the pump pulses were unchanged. In this implementation, both τ$_2$ and τ$_1$ typically ranged between 4 and 7 μs for shorter and longer distances, respectively.

The data were analysed using DeerAnalysis[77], and distance distributions were obtained using Tikhonov regularisation. The regularisation parameter was chosen either by the L-curve criterion or by visual inspection in order to obtain reasonable fits. Estimation of uncertainties in distance distributions due to background correction and noise were obtained using the validation option in DeerAnalysis[77].

**Crystallisation of AncCDT-5 and structure determination**. A solution of TEV-cleaved AncCDT-5 (expressed in BL21(DE3) cells and purified by IMAC as described above) was further purified by SEC using a HiLoad 16/600 Superdex 75 pg column (GE Healthcare, USA) in 50 mM Tris-HCl, pH 7.5, 20 mM NaCl, and was concentrated to 30 mg mL$^{-1}$ using an Amicon Ultra-15 filter unit (10 kDa MWCO). The protein was crystallised using the vapour diffusion method at 18 °C. Crystallisation screens were set up using the PACT premier HT-96 C10 screen from Molecular Dimensions (Newmarket, USA) with drops containing 200 nL reservoir solution and 200 nL AncCDT-5 solution. AncCDT-5 crystals formed in 28 days in 20% w/v PEG 6000, 0.2 M $MgCl_2$, 0.1 M HEPES, pH 7.0.

Crystals were cryoprotected (30% w/v PEG 6000, 0.2 M $MgCl_2$, 0.1 M HEPES pH 7.0) and frozen in liquid nitrogen. Data were collected at 100 K on the MX2 beamline of the Australian Synchrotron. The data were indexed and integrated in XDS[78] and scaled in AIMLESS[79]. The structure was solved by molecular replacement in Phaser[80] using the two domains of *Pa*CDT as search models (PDB 3KBR small domain = residues 122–223; large domain = residues 27–121 and 224–258). The structure was refined by real-space refinement in Coot[81] and using

REFMAC[82] and phenix.refine[83]. Data collection and refinement statistics are given in Supplementary Table 1. Residues in the AncCDT-5 structure were numbered according to the equivalent positions in AncCDT-1 (PDB 5TUJ)[40] and the coordinates deposited in the Worldwide Protein Data Bank (PDB 6WUP).

**Molecular dynamics (MD) simulations**. 500 ns simulations were performed in Desmond[84] (in Schrödinger 2019-11) using the OPLS3e force field[85]. These simulations were initiated from the *Pa*CDT–acetate trimer (PDB 5HPQ, residues 13–250 in PDB, $n = 1$), unliganded AncCDT-1 (PDB 5T0W, chain A, residues 14–246, $n = 3$), AncCDT-3/L188 (PDB 5JOS, residues 12–247, $n = 3$), AncCDT-3/P188 ($n = 3$) and AncCDT-5 (PDB 6WUP, residues 13–245, $n = 3$) structures, with all small molecules removed. The AncCDT-3/P188 structure was obtained by making the L188P mutation and local minimisation in Maestro (Schrödinger Release 2019-1: Maestro, Schrödinger, LLC, New York, NY, 2019). Desmond was used to add N-terminal acetyl caps and C-terminal amide caps to each structure and for energy minimisation of the protein structures. Each protein was solvated in an orthorhombic box (15 Å buffer periodic boundary) with SPC water molecules. The smallest number of counter ions ($Na^+$ or $Cl^-$) needed to neutralise the net charge of each system were added. Energy minimisation was achieved using a hybrid method of the steepest descent algorithm and the limited-memory Broyden–Fletcher–Goldfarb–Shanno algorithm (maximum of 2000 iterations and a convergence threshold of 1 kcal $mol^{-1}$ $Å^{-1}$). The system was relaxed using the default relaxation procedure in Desmond. For production MD simulations of the NPT ensemble, the temperature was maintained at 300 K using a Nosé–Hoover thermostat (relaxation time = 1.0 ps) and the pressure was maintained at 1.01 bar (relaxation time = 2.0 ps) using a Martyna–Tobias–Klein barostat. Otherwise, default Desmond options were used. Following relaxation of the system, each simulation was run for 500 ns. Distances between the α-carbons of the tagged residues and the radii of gyration were calculated using the ProDy package on snapshots sampled every 0.1 ns[86]. In addition, representative snapshots (with a range of Cα–Cα distances) corresponding to the closed, open and wide-open states (if sampled) were selected for modelling of Gd(III)–Gd(III) distances. Allocation of states (closed/open/wide-open) was done based on Cα–Cα measurements. For AncCDT-1, conformations with Cα–Cα distances between ~3.3 and 3.7 nm could be separated into the open and wide-open conformations by principal component analysis. Plots of Cα–Cα distances over the course of each simulation are provided in Supplementary Fig. 10.

**Modelling Gd(III)–Gd(III) distances**. To calculate predicted Gd(III)–Gd(III) distances of different conformations, propargyl-DO3A-Gd(III) tags were modelled onto the crystal structures of AncCDT-1 (PDB 5T0W, 5TUJ), AncCDT-3/L188 (PDB 5JOS), AncCDT-5 (PDB 6WUP), *Pa*CDT (PDB 3KBR, 5HPQ) and selected MD snapshots. To do this, the mutation tool in PyMOL was used to identify the side-chain dihedral angles $\chi_1$ and $\chi_2$ of a phenylalanine residue that produced the least amount of steric clashes with the rest of the protein; these angles were used when modelling the tag. The angles $\chi_9$ and $\chi_{10}$ (Supplementary Fig. 3) were fixed to $-140°$ and $70°$, respectively, to allow coordination of the metal ion by the nearest nitrogen of the triazole ring. The angle $\chi_6$ was set to $180°$. As in previous work with a closely related cyclen tag[51], setting $\chi_6$ to $180°$ (Supplementary Fig. 3c) yielded a better correlation between experimental and back-calculated Gd(III)–Gd(III) distances compared with setting $\chi_6$ to $0°$ (Supplementary Fig. 3d). Adding additional flexibility to the model by allowing rotation around $\chi_9$ (any angle) and $\chi_{10}$ ($-60°$, $60°$ or $180°$) increased the width of modelled distance distributions but did not significantly change the positions of the peaks within these distributions, which still correlated well with experimental measurements (Supplementary Fig. 3e) and the distances predicted using the single conformation approach mentioned above.

**Reporting summary**. Further information on experimental design is available in the Nature Research Reporting Summary linked to this paper.

## Data availability

The atomic coordinates and structure factors for AncCDT-5 (PDB 6WUP) have been deposited in the Worldwide Protein Data Bank (http://wwpdb.org/). Any other relevant data are available from the corresponding authors upon reasonable request. Source data are provided with this paper.

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

## Acknowledgements

J.A.K. acknowledges financial support from an Australian Government Research Training Program Scholarship. B.E.C. acknowledges financial support from a Rod Rickards PhD Scholarship. Funding by the Australian Research Council, including a Laureate Fellowship to G.O., is gratefully acknowledged. D.G. acknowledges the support of the Minerva Foundation, and this research was made possible in part by the historic generosity of the Harold Perlman Family (D.G.). D.G. holds the Erich Klieger Professorial Chair in Chemical Physics. We thank Yasmin Ben-Ishay for her help with sample preparation and DEER measurements.

## Author contributions

J.A.K. performed prephenate dehydratase assays (including protein purification), ran SEC-MALS experiments, processed crystallographic data, solved the structure of AncCDT-5 and performed molecular dynamics simulations. M.C.M. selected tagging sites, performed cloning of genes into pETMCSIII, expressed and purified protein for crystallography and DEER experiments, tagged these variants, modelled Gd(III)–Gd(III) distances and crystallised AncCDT-5. A.F. performed DEER measurements on the samples and analysed the DEER data. B.E.C. performed essential preliminary work and guided the direction of this project. L.A.A. prepared the propargyl-DO3A tags. J.A.K., M.C.M, A.F., D.G., G.O. and C.J.J. analysed the data and wrote the paper. J.A.K., M.C.M. & A.F. produced the figures. D.G., G.O. and C.J.J. supervised the project. All authors reviewed the paper.

## Competing interests

The authors declare no competing interests.
