## [Peer Review File · Nature Communications]

Reviewers' comments:

Reviewer #1 (Remarks to the Author):

Kaczmarek et al. provide an interesting manuscript exploring an integrated structural biology approach aiming to understand the different catalytic activities of the ancestors of enzymes evolved from SBPs. The interesting point is that the binding sites are similar but the proteins sample different conformations and that the difference in activities is rooted in the protein conformational landscape rather than differences in their active sites.

Your interesting finding and the combination of crystallography, MD and DEER EPR with a new cysteine independent labeling strategy is a significant new piece of methodology that clearly merits publication. Nevertheless, there are several points that require clarification or expansion to make this work rigorous and ensure the highest level of reproducibility.

1. Your entire interpretation of the DEER data critically depends on modelling the label into Xray structures or MD snapshots. You arbitrarily choose some dihedrals over others and completely ignore the different conformers that will likely be populated. I appreciate that MTSWizard or MMM might not include tagged AzF but without at least showing the results of $\chi^2=0$ you have just chosen to model in a way that the outcome suits your expectation. This is not rigorous.

2. Your DEER data analysis is not described sufficiently.

(i) Your background correction is not always 3D homogeneous as expected for soluble proteins, how do you justify this?

(ii) Some of your raw data seems to have blips and these have been cut in processing. Are these instrumental artefacts?

(iii) You use the validation tool but do not describe the validation parameters. You need to describe this in detail. If you only explored a narrow range of parameters, the error bars will be small but meaningless. You need to add appropriate error bars/confidence intervals to the figures in the main text, they are barely visible in the SI.

(iv) DeerAnalysis provided you with color-coded reliabilities for different ranges in your distance distribution that you deliberately choose not to reproduce here. Why did you choose to remove this? For AncCDT-3/P188 and AncCDT-5 the maxima of the distributions lie in regions where the shape of the distribution is not reliable anymore.

3. Your discussion of binding and multimerization assumes the binding constants at room temperature and under cryogenic EPR conditions are similar. What is your justification for this?

4. You only observe a substrate induced conformational change in AncCDT-1 +/- L-Arg. Here, only the refolded sample -L-Arg shows a different distribution. How many replicates were made of this? For both AncCDT-3/L188 and AncCDT-5 you only observe the open conformation in DEER but crystallize different conformations. What attempts did you make to observe the change in DEER? The presence of a wide-open state may be extrapolated from the MD but is not backed by DEER.

5. Your interpretation of the C DEER data is that the protein is sweeping a range of conformations while being frozen and trapped in several states leading to a broad distribution. All other constructs freeze in an open or bimodal conformation. You attribute this to the different populations of states. I tend to disagree: Assuming that the cryogenic observations are relevant in solution I would think that not the different populations (determined by the relative energies) but the barrier for the conformational change is the main difference. It seems low enough in PaCDT that a variety of conformers is frozen out in the DEER sample rather than funneling into two states as in AncCDT-1 - L-Arg. A discussion of this might benefit the readers.

Minor

-You cite some of your own work for orientation selection and multi-spin effects. It is okay to cite

your work, but your work was not the first on these topics.

-When estimating labeling by mass spec (the data seems to be missing), how can you be sure both tagged and non-tagged protein give equally strong peaks?

-At 10-fold dilution presuming the tag does not perturb multimerization, the probability of having one, two or three tags per trimer will be 24.3%, 2.7% and 0.1 %

Reviewer #2 (Remarks to the Author):

In this manuscript, Kaczmarski, et al. report a study of a series of proteins along the evolutionary path from arginine SBP to cyclohexadienyl dehydratase combining data from DEER measurements, X-ray crystallography and MD simulations. The study is focused on the comparison of conformational dynamics of these proteins. In the end, it was proposed that the protein conformational dynamics was gradually changed by mutations distant from the active site to eliminate the nonfunctional state. Such a focused effort using a diverse array of different investigative techniques to study a single biochemical function is commendable. The results were well presented.

However, I have a few concerns about the data analysis and interpretation:

1. Although the authors argued that the wide-open conformation might be accessible for AncCDT-1, the DEER data at least demonstrate that the population is much smaller than those in AncCDT-3 and AncCDT-5. How do we understand the abundance of this state in AncCDT-3 and AncCDT-5 with respect to AncCDT-1 from the angle of evolution?
2. Could the authors explain how the mutations distant from active site favor the oligerization of PaCDT?
3. To my understanding, for each protein system three 500 ns MD simulations were conducted. Are the data in Figure 3 the collection of all three trajectories? Did the three simulations sample the similar conformational spaces? It would be better to show the time evolution of all three trajectories, i.e. the Ca-Ca distance variations along the three trajectories.
4. Page 9, the authors mentioned principle component analysis, but it was not clearly indicated why the authors use two methods to allocate the conformational states.
5. Page 9, "...and the systems were neutralized using Na⁺ or Cl⁻ ions." What is the concentration of NaCl in the simulation system?
6. Page 17, Line 497, '1.5 ms' should be '1.5 μs'.

Reviewer #3 (Remarks to the Author):

Kaczmarski et al. report an exhaustive analysis into the evolutionary emergence of enzyme catalysis from substrate binding. The work clearly demonstrates that the enhancement of a new catalytic function is not simply a matter of improving the catalytic machinery at the active site, but optimisation of conformational dynamics through distant mutations can also play a prominent role. The work should be of wide interest to researches working in molecular evolution and enzyme engineering. I have, however, a number of suggestions for major revision, some of them related with the need to emphasise novel aspects of the work which, in my view, are not made sufficiently clear to the reader:

The authors claim in the the discussion section that this is the first time the propargyl-DO3A-GD(III)/DEER approach is applied to study rigid body protein dynamics. This first, however, is not mentioned in the abstract. The authors should explain convincingly the relevance and implications of this new possibility, providing some general background of the methodology and discussing why they think the first application reported provides a basis for the characterisation of protein

dynamics in many different systems, i.e., the authors should explain why the new methodology is important beyond the specific protein system characterised in the submitted manuscript.

The abstract seems to imply that remote mutations along the evolutionary trajectory from the ancestral protein to the modern enzyme have frozen out non-productive conformations and led to the sampling of compact states that are catalytically competent. However, we learn in the discussion section that the situation is more complex, as the modern enzyme is actually sampling different conformations and such sampling is relevant for function, as a closed state is optimised for catalysis, while population of a moderately open state is required for binding and product release. This is clearly different than the simpler view seemingly conveyed in the abstract. The nature of the evolutionary change in conformational dynamics and its relation with function should be discussed in a more detailed and convincing manner and, perhaps, the abstract will need to be modified accordingly.

Solute binding proteins are peculiar, in the sense that they can populate substantially different conformations and such conformational diversity can be repurposed during evolution to lead to a new function, as the authors show. The issue arises, therefore, of the generality of the characterised mechanism for enzyme emergence and evolution. This needs to be discussed in some detail. Would the evolutionary mechanism based on repurposing conformational diversity apply to other protein systems (not only solute binding proteins)? Is there literature evidence for the proposed mechanism with other protein systems?. How common can we expect evolution of new functions through remote mutations to be?. Again, is there previous reported evidence in the literature with other systems of such evolutionary role of remote mutations?

Response to Reviewers' Comments

Reviewer 1:

1. Your entire interpretation of the DEER data critically depends on modelling the label into Xray structures or MD snapshots. You arbitrarily choose some dihedrals over others and completely ignore the different conformers that will likely be populated. I appreciate that MTSLWizard or MMM might not include tagged AzF but without at least showing the results of $\chi_6=0$ you have just chosen to model in a way that the outcome suits your expectation. This is not rigorous.

We have made changes to the manuscript to address this comment. In particular, we now show a comparison of the correlations between modelled and experimental DEER distances when we model with $\chi_6=0^\circ$ (see **Supplementary Figure 3C**) and $\chi_6=180^\circ$ (**Supplementary Figure 3D**), respectively. These plots illustrate the better correlation between experimental and predicted Gd-Gd distances when $\chi_6=180^\circ$, as observed previously for a similar Gd(III) tag with modified pendants (reference 4 of the SI, (Abdelkader et al., 2015)). In addition, we simulated distance distributions with $\chi_6=180^\circ$, free rotation around χ_9 , and allowing χ_{10} to be either -60° , 60° or 180° (**Supplementary Figure 3E**).

2. Your DEER data analysis is not described sufficiently. (i) Your background correction is not always 3D homogeneous as expected for soluble proteins, how do you justify this?

We have made changes to the manuscript to address this comment. In all cases, we chose the background dimensionality according to the best fit procedure in DeerAnalysis. In most cases the background dimensionality was either 3 or showed only a small deviation from $D=3$. In our experience, deviations away from $D=3$ can occur for Gd(III) spins labels especially when the evolution time is long and when we apply chirp pump pulses (as is the case for the approach used here). This deviation of the background might be due to the high spin of Gd(III) and is currently being investigated .

The main outlier was the trace presented in **Supplementary Figure 7D** (the diluted *Pa*CDT sample), which showed a poor fit when $D=3$ (shown in **Panel A** of the figure below). Obviously, the fit with $D=3$ could have been improved if we cut the last 1 μ s of the trace (**Panel B** of the figure below), but then this would have been at the expense of accurate determination of the background decay. In the case of the *Pa*CDT trace in **Supplementary Figure 7D**, we found the fit could be significantly improved by setting $D=4$ (a copy of this data is shown in **Panel C**, below, for your reference). In addition, as part of the validation process in DeerAnalysis, the values of D and the start and end points of the background correction are varied slightly – this is taken into account

when calculating the striped region (confidence intervals) shown in these plots. Allowing variation in the background dimension is justified because the trace is long enough to allow for good estimation of the background and (ii) the fit to the data is improved without significantly changing the resulting distance distributions (especially the position of the peaks of the distance distributions).

Data for review: DEER data for diluted sample of PaCDT, comparing the quality of fits and associated Gd-Gd distance distributions when (A) D=3 and 5.5 μs of the trace is used for the fit, (B) D=3 but only the first 4 μs of the trace is used for the fit, and (C) D=4, with 5.5 μs of the trace used for the fit. The data shown in (C) is a copy of the data presented in Figure 3 and Supplementary Figure 7.

We added in the caption of **Supplementary Figure 4** "The parameter ranges used for the validations are: white noise 0-1.5, background start $0.2 \cdot t_{\text{max}} - 0.6 \cdot t_{\text{max}}$ and background dimension 3-3.6. The limited variation in background dimension was permitted, as a non-homogeneous background improved the fit for some of the traces." Similar details are also provided in the captions of **Supplementary Figure 6** and **Supplementary Figure 7**.

(ii) Some of your raw data seems to have blips and these have been cut in processing. Are these instrumental artefacts?

We have made changes to the manuscript to address this comment. These are instrumental artefacts and were not taken into account in the analysis. This is now stated clearly in the captions of **Supplementary Figures 4, 6 & 7**.

(iii) You use the validation tool but do not describe the validation parameters. You need to describe this in detail. If you only explored a narrow range of parameters, the error bars will be small but meaningless. You need to add appropriate error bars/confidence intervals to the figures in the main text, they are barely visible in the SI.

We have made changes to the manuscript to address this comment. We repeated the analysis with a broader range of parameters and remade **Supplementary Figures 4, 6 & 7**. As stated above, we added in the caption of **Supplementary Figure 4** "The parameter ranges used for the validations are: white noise 0-1.5, background start $0.2 \cdot t_{\max} - 0.6 \cdot t_{\max}$, and background dimension 3-3.6. The limited variation in background dimension was permitted, as a non-homogeneous background improved the fit for some of the traces." We also refer to these in the captions of **Supplementary Figures 5, 6 & 7**.

We prefer not to add anything to **Figure 3** as it is already pretty crowded. The confidence intervals associated with the data in **Figure 3** are shown in **Supplementary Figures 4, 6 & 7**, and we feel this suffices. We have, however, remade **Figure 3** to show the distance distributions that were obtained using the broader range of validation parameters mentioned above.

(iv) DeerAnalysis provided you with color-coded reliabilities for different ranges in your distance distribution that you deliberately choose not to reproduce here. Why did you choose to remove this? For AncCDT-3/P188 and AncCDT-5 the maxima of the distributions lie in regions where the shape of the distribution is not reliable anymore.

We have made changes to the manuscript to address this comment. In particular, we have added color-coded reliabilities to all the figures in the SI where relevant. As the reviewer points out, the maxima of the distance distributions of AncCDT-3/P188 and AncCDT-5 are in regions where the *shape* of the distribution is not reliable. However, the *mean distances* are still reliable within these regions. We did not use or interpret the shape of the distance distributions of these variants anywhere in our conclusions, i.e. only the maxima of the mean distance distributions were considered (which are reliable).

3. Your discussion of binding and multimerization assumes the binding constants at room temperature and under cryogenic EPR conditions are similar. What is your justification for this?

We have made changes to the manuscript to address this comment. The temperature that one has to consider regarding binding and multimerization in the EPR samples is the temperature at which the samples were equilibrated before freezing (now stated more clearly on p. 20). The samples were flash frozen by insertion into the cold (10 K) magnet bore (now stated on p. 24), so it is reasonable to assume they retained the equilibrium populations pertaining to the freezing temperature of 80:20 (v/v) water/glycerol, which is about -6 °C (Lane, 1925). For a discussion of this effect, we added the following sentences on p. 20: "Considering that DEER data are recorded of frozen solutions, it cannot be ruled out that conformational equilibria established at room temperature change during snap-freezing of the samples. Notably, however, the freezing point of 20 % deuterated glycerol solutions is only a little below 0 °C (Lane, 1925), suggesting that any changes in conformational equilibria due to freezing would be minor."

4.(a) You only observe a substrate induced conformational change in AncCDT-1 +/- L-Arg. Here, only the refolded sample -L-Arg shows a different distribution. How many replicates were made of this?

We have made changes to the manuscript to address these comments. To address the first question (about the ligand induced changes in conformational distribution of AncCDT-1), we only observe a ligand-induced conformational change in the refolded sample of AncCDT-1 since the natively purified sample of AncCDT-1 co-purifies with ligand bound from the expression system - this is typical of amino acid-binding proteins (Clifton et al., 2018; Clifton and Jackson, 2016) and is explained on page 10 of the manuscript. We performed the same experiment with the native *Salmonella enterica* lysine-arginine-ornithine-binding protein (SeLAOBP) in the absence and presence of L-Arg (presented in **Supplementary Figure 5**), which exhibited the same behaviour as AncCDT-1. We now write on page 10: "A ligand-induced shift towards a more closed state was also observed when L-arginine was added to a sample of a lysine-arginine-ornithine binding protein from *Salmonella enterica* (SeLAOBP, **Supplementary Figure 5**)."

(b) For both AncCDT-3/L188 and AncCDT-5 you only observe the open conformation in DEER but crystallize different conformations. What attempts did you make to observe the change in DEER?

In regards to the other proteins and the wide-open state, the DEER data clearly show that AncCDT-3 and AncCDT-5 favour the wide-open state in solution: the dominant peaks in the DEER-measured

Gd-Gd distance distributions for these samples are consistent with the predicted Gd-Gd distances from crystal structures and MD snapshots of the wide-open states of these proteins. For AncCDT-3, the peak in the DEER measured distance distribution is comparable to that predicted from the wide-open crystal structure of AncCDT-3/L188. This wide-open conformation is also stable for several hundred ns in each MD simulation of AncCDT-3. Therefore, the DEER, crystallography and MD data support each other for this protein.

In contrast, while DEER data show that AncCDT-5 favours the wide-open conformation in solution, we do not have a crystal structure of the open or wide-open conformations of this protein. Instead, our crystal structure of AncCDT-5 is stabilized in the closed conformation, with a molecule of HEPES (present at 0.1 M) stabilizing the closed state. However, our MD simulations of AncCDT-5 do show how this structure rapidly opened towards the wide-open state when this HEPES molecule was removed. To test whether the opposite (i.e. open-to-closed) shift could be induced in solution, we did perform a DEER experiment on AncCDT-5 in the presence of 500 mM HEPES (well above physiological relevance, e.g. for AncCDT-1, $K_d = 0.32 \mu\text{M}$ for L-Arg). While this data showed a small overall shift towards more closed states in the presence of HEPES, the concentration of HEPES used in these tests was not sufficient to significantly stabilize the closed conformation. It is likely that the closed state in the crystal structure is further stabilized by the crystal lattice, macromolecular crowding from 20% PEG 6000 and very high protein concentration (30 mg/mL) vs. the relatively dilute conditions in the DEER experiments (100 μM AncCDT-5). In other words, we could not replicate the conditions in the crystal structure (crystal lattice, crowding, concentration). However, we feel that the data we have obtained by DEER, crystallography and MD are robust and do paint a fairly clear picture of a protein that can fluctuate between open and closed states, with the close state being the preferred state in solution.

Data for review: AncCDT-5 in the presence of 500 mM HEPES.

Performing DEER experiments on AncCDT-3, AncCDT-5 or PaCDT in the presence of the enzymatic substrates was not feasible due to the rapid turn-over of the substrates (e.g. $k_{\text{cat}} \sim 18 \text{ s}^{-1}$ for

PaCDT), and no other ligands have been identified that could have been used to stabilize these proteins in the closed state.

(c) The presence of a wide-open state may be extrapolated from the MD but is not backed by DEER. We assume this is referring to AncCDT-1 (the wide-open states are clearly dominant in AncCDT-3 and AncCDT-5 in the DEER data). The only ancestral variant where we don't have unambiguous experimental support (either X-ray or DEER) for the wide-open state is AncCDT-1. However, for this variant the wide-open state is spontaneously sampled by MD and is stable for several hundred ns (using one of the most advanced and reliable force fields available, the all atomistic force field OPLS3e (Roos et al., 2019)), it is unambiguously observed in crystal structures of close relatives (e.g. ArgBP from *Thermotoga maritima*, PDB 4PRS (Ruggiero et al., 2014)) and, most importantly, sampling of this state is not *excluded* by the DEER data. As shown in **Figure 3**, the predicted signals from the wide-open state and the open state overlap (they are too close to produce resolved separate peaks) and are both consistent with the peak from 3.9 – 5.0 nm shown in **Figure 3**. As shown in **Figure 3**, the inter-tag distance from the crystal structure of the open state is located at the lower bounds of this peak (unlike the closed state that is located in the centre of the closed peak) at 4.1 Å, so there is every chance that a proportion of this peak is a result of the wide-open state. So, we have the most conservative interpretation of the data – that it most likely is present – however, we have tried to be cautious in our description and interpretation (e.g. in our discussion on **pages 11-12 and 19-20**).

5. Your interpretation of the C DEER data is that the protein is sweeping a range of conformations while being frozen and trapped in several states leading to a broad distribution. All other constructs freeze in an open or bimodal conformation. You attribute this to the different populations of states. I tend to disagree: Assuming that the cryogenic observations are relevant in solution I would think that not the different populations (determined by the relative energies) but the barrier for the conformational change is the main difference. It seems low enough in PaCDT that a variety of conformers is frozen out in the DEER sample rather than funneling into two states as in AncCDT-1 - L-Arg. A discussion of this might benefit the readers.

We have made changes to the manuscript to address these comments. We now write on **page 20**: "Considering that DEER data are recorded of frozen solutions, it cannot be ruled out that conformational equilibria established at room temperature change during snap-freezing of the samples. Notably, however, the freezing point of 20 % deuterated glycerol solutions is only a little

below 0 °C (Lane, 1925), suggesting that any changes in conformational equilibria due to freezing would be minor. It is thus significant that the energy landscape of the extant enzyme *Pa*CDT led to the freezing out of a continuous range of conformations, whereas that of the amino-acid binding protein AncCDT-1 funnelled the protein into two clearly distinct conformations. Importantly, the distances measured by DEER here are in good agreement with distances predicted from the crystal structures and states obtained from MD simulations.”

Minor

-You cite some of your own work for orientation selection and multi-spin effects. It is okay to cite your work, but your work was not the first on these topics.

We added four additional references on p. 4.

-When estimating labeling by mass spec (the data seems to be missing), how can you be sure both tagged and non-tagged protein give equally strong peaks?

Peak heights in mass spectrometry are well-known to depend on the proteins' capacity to transition into the vacuum phase. Therefore, we report only the indicative tagging yields and the method used to determine them (i.e. mass spectrometry) on p. 24.

-At 10-fold dilution presuming the tag does not perturb multimerization, the probability of having one, two or three tags per trimer will be 24.3%, 2.7% and 0.1 %

The reviewer is right. We have corrected the numbers on p. 14.

Reviewer #2 (Remarks to the Author):

1. Although the authors argued that the wide-open conformation might be accessible for AncCDT-1, the DEER data at least demonstrate that the population is much smaller than those in AncCDT-3 and AncCDT-5. How do we understand the abundance of this state in AncCDT-3 and AncCDT-5 with respect to AncCDT-1 from the angle of evolution?

We have made changes to the manuscript to address these comments. We now write on page 20:

“The role of the wide-open state along the evolutionary trajectory from AncCDT-1 to *Pa*CDT is more difficult to interpret. We observe this state in crystal structures of extant relatives of AncCDT-1, it is stable for several hundred ns in MD simulations of AncCDT-1 and (because of the overlap of the DEER signals between the wide-open and open states) it is also consistent with the DEER data. However, it is clear that there was a shift towards the wide-open state after AncCDT-1 (as seen in

AncCDT-3 and AncCDT-5). In our previous work (Clifton et al., *Nature Chem. Biol.* 2018), we were unable to identify a natural ligand for the non-enzymatic binding protein AncCDT-2 (which predates AncCDT-3 and AncCDT-5); although it had lost its ancestral arginine binding function, it (and its decedents such as Pu1068 from the aquatic bacterium *Pelagibacter ubique*) retain the genomic organization and biophysical properties of solute binding proteins. Thus, we speculate that the shift towards the wide-open state in the intermediate ancestors could have resulted from the selective pressure for the (still unknown) solute-binding function that preceded the evolution of CDT activity.”

Figure for reviewers: Representation of the phylogenetic tree showing the position of the clade containing binding proteins of unknown function (including Pu1068 from *Pelagibacter ubique*), which are the decedents of the intermediate ancestor, AncCDT-2 (unknown solute binding function).

2. Could the authors explain how the mutations distant from active site favor the oligerization of PaCDT?

We have made changes to the manuscript to address these comments. In the trimeric crystal structure of PaCDT (e.g. PDB 5HPQ), the interface between neighbouring subunits (i.e. chain A and chain B) is formed through numerous interactions (primarily between the large domains of the proteins). In 5HPQ, for example, the interface between neighbouring subunits leads to a buried surface area of approximately 745 Å² and features a number of salt bridges and the complementary packing of hydrophobic residues. Of the 31 residues that contribute to this interface 19 positions (i.e. >60 %) are substituted between AncCDT-5 and PaCDT. Mutations on the branch separating these two proteins introduce several key interactions that would stabilize the trimeric form. For example

- T97R(A) – M221Q(B) introduces salt bridge between neighbouring subunits,
- L217H(B) introduces salt bridge with Glu197(A),
- Substitutions including H195F(A), H218I(B) and P83L(A) improve hydrophobic packing at the interface.

Additional substitutions at and near the interface also likely contribute to the stabilization of the trimeric form.

We have added the following to p. 16:

“A number of key residues that contribute to the inter-subunit interfaces in the crystal structures of *Pa*CDT are not found in *Anc*CDT-5, explaining this difference in oligomeric states. For example, the substitutions T97R, M221Q and L217H introduce specific electrostatic interactions between chains in *Pa*CDT, while H195F, H218I and P83L appear to improve packing between the subunits. However, oligomerization is a complex process and remote and unpredictable mutations can also have substantial influence on oligomeric states (Fraser et al., 2016). Accordingly, we cannot exclude the possibility that additional mutations remote from the interface also play a role.”

3. To my understanding, for each protein system three 500 ns MD simulations were conducted. Are the data in Figure 3 the collection of all three trajectories? Did the three simulations sample the similar conformational spaces? It would be better to show the time evolution of all three trajectories, i.e. the Ca-Ca distance variations along the three trajectories.

We have made changes to the manuscript to address these comments. The reviewer is correct in thinking that three separate 500 ns MD simulations were conducted for each system of *Anc*CDT-1, *Anc*CDT-3/P188, *Anc*CDT-3/L188 and *Anc*CDT-5. We have added plots showing the $C\alpha$ - $C\alpha$ distance variations along the three trajectories in **Supplementary Figure 9**. These highlight that in each of the three simulations of each system sampled mostly similar conformational spaces during the 500 ns of simulation. One of the *Anc*CDT-1 MD simulations did sample somewhat different conformational space compared to the other two (i.e. sampling more of the wide-open conformation)– this difference is likely due to there being only a small energy gap between the open and wide-open conformational states, and the fact that different seed values were used to determine initial velocities in each of the three replicates. However, it is notable that it was stable in this wide-open state for several hundred ns, suggesting that it is a relatively low-energy state.

4. Page 9, the authors mentioned principle component analysis, but it was not clearly indicated why the authors use two methods to allocate the conformational states.

We have made changes to the manuscript to address this comment. Typically, the C α -C α distance was used to group conformational snapshots into closed, open and wide-open states and did not require more complex analysis. As seen in **Figure 4**, the C α -C α distance showed a good correlation with the radius of gyration of these proteins (reflecting the extent of their opening). However, in the case of AncCDT-1, it can be seen that there are in fact two semi-discrete populations that sample C α -C α distances from ~3.3-3.7 nm, i.e. the overlap of the most open range of the open state and the most closed range of the wide-open state. In this case, a more sophisticated approach was required to separate these populations into the open and wide-open populations as described on **p. 26**.

The text on **page 26** now reads:

“For AncCDT-1, conformations with C α -C α distances between ~3.3 and 3.7 nm could be separated into the open and wide-open conformations by principle component analysis.”

5. Page 9, “...and the systems were neutralized using Na⁺ or Cl⁻ ions.” What is the concentration of NaCl in the simulation system?

Other than the Na⁺ and Cl⁻ ions used to neutralize the net charge of the system, no additional NaCl was included in the simulation systems. We have now clarified this in the text on **page 26**: “The smallest number of counter ions (Na⁺ or Cl⁻) needed to neutralize the net charge of each system were added.”

6. Page 17, Line 497, ‘1.5 ms’ should be ‘1.5 μ s’.

Thank you for pointing out this typo. We have fixed this to read “1.5 μ s” (now on **page 13**).

Reviewer #3 (Remarks to the Author):

The authors claim in the discussion section that this is the first time the propargyl-DO₃A-GD(III)/DEER approach is applied to study rigid body protein dynamics. This first, however, is not mentioned in the abstract. The authors should explain convincingly the relevance and implications of this new possibility, providing some general background of the methodology and discussing why they think the first application reported provides a basis for the characterisation of protein dynamics in many different systems, i.e., the authors should explain why the new methodology is important beyond the specific protein system characterised in the submitted manuscript.

We have made changes to the manuscript to address this comment. We have added the following highlighted section to the abstract (we are not allowed to make claims around novelty or “first”, etc. owing to editorial policy):

Several enzymes are known to have evolved from non-catalytic proteins such as solute-binding proteins (SBPs). Although attention has been focused on how a binding site can evolve to become catalytic, an equally important question is: how do the structural dynamics of a binding protein change as it becomes an efficient enzyme? Here we performed a variety of experiments, including propargyl-DO₃A-GD(III) tagging and double electron-electron resonance (DEER) to study the rigid body protein dynamics of reconstructed evolutionary intermediates to determine how the conformational sampling of a protein changes along an evolutionary trajectory linking an arginine SBP to a cyclohexadienyl dehydratase (CDT). We observed that primitive dehydratases predominantly populate catalytically unproductive conformations that are vestiges of their ancestral SBP function. Non-productive conformational states, including a wide-open state, are frozen out of the conformational landscape via remote mutations, eventually leading to extant CDT that exclusively samples catalytically relevant compact states. These results show that remote mutations can reshape the global conformational landscape of an enzyme as a mechanism for increasing catalytic activity.

The abstract seems to imply that remote mutations along the evolutionary trajectory from the ancestral protein to the modern enzyme have frozen out non-productive conformations and led to the sampling of compact states that are catalytically competent. However, we learn in the discussion section that the situation is more complex, as the modern enzyme is actually sampling different conformations and such sampling is relevant for function, as a closed state is optimised for catalysis, while population of a moderately open state is required for binding and product release. This is clearly different than the simpler view seemingly conveyed in the abstract. The nature of the evolutionary change in conformational dynamics and its relation with function should be discussed in a more detailed and convincing manner and, perhaps, the abstract will need to be modified accordingly.

We have made changes to the manuscript to address these comments. First, we have modified the abstract to point out that the wide-open state is depopulated in the extant enzyme *Pa*CDT. “Non-productive conformational states, including a wide-open state, are frozen out of the conformational landscape via remote mutations”. Unfortunately, with the 150 word limit (we are already at 169 words), this is as much as we can do in terms of adding new information to the abstract.

Second, on **page 20**, we expand the discussion, which the reviewer is correct in stating appears to be more complex than just freezing out a non-productive state. This overlaps with our response to reviewer 2, point 1. It's an interesting question, but one difficulty we faced is we have not been able to identify the natural ligand for the intermediate clade of solute binding proteins (we spent several years trying several hundred ligands), and their ancestor AncCDT-2. The editor and reviewers of that Nature Chem. Biol. paper accepted that this orphan function did not need to be known precisely for the conclusions regarding the evolution of enzymatic activity to be made. So, while it seems to us to be logical that there was selection pressure to favour the wide-open state, before the transition to CDT activity (thus explaining the dominance of this state in AncCDT-2>5) we wanted to be as conservative as possible. We now write:

"The role of the wide-open state along the evolutionary trajectory from AncCDT-1 to PaCDT is more difficult to interpret. We observe this state in crystal structures of extant relatives of AncCDT-1, it is stable for several hundred ns in MD simulations of AncCDT-1 and (because of the overlap of the DEER signals between the wide-open and open states) it is also consistent with the DEER data. However, it is clear that there was a shift towards the wide-open state after AncCDT-1 (as seen in AncCDT-3 and AncCDT-5). In our previous work (Clifton et al., Nature Chem. Biol. 2018), we were unable to identify a natural ligand for the non-enzymatic binding protein AncCDT-2 (which predates AncCDT-3 and AncCDT-5); although it had lost its ancestral arginine binding function, it (and its decedents such as Pu1068 from the aquatic bacterium *Pelagibacter ubique*) retain the genomic organization and biophysical properties of solute binding proteins. Thus, we speculate that the shift towards the wide-open state in the intermediate ancestors could have resulted from the selective pressure for the (still unknown) solute-binding function that preceded the evolution of CDT activity."

Solute binding proteins are peculiar, in the sense that they can populate substantially different conformations and such conformational diversity can be repurposed during evolution to lead to a new function, as the authors show. The issue arises, therefore, of the generality of the characterised mechanism for enzyme emergence and evolution. This needs to be discussed in some detail. Would the evolutionary mechanism based on repurposing conformational diversity apply to other protein systems (not only solute binding proteins)? Is there literature evidence for the proposed mechanism with other protein systems? How common can we expect evolution of new functions through

remote mutations to be? Again, is there previous reported evidence in the literature with other systems of such evolutionary role of remote mutations?

These are good suggestions, questions and comments. We have made changes to the manuscript to address these comments. We have modified the discussion to provide more context in terms of how general this mechanism of functional change via altered conformational sampling might be on **page 20**:

“The findings from this work add to growing evidence that the modification of protein dynamics often plays an important role in the evolution of new protein function. While the current work provides a unique example for pronounced, historical conformational changes between a non-catalytic ancestral protein and an efficient enzyme, our findings mirror those of other work that highlight how catalytic activity can be enhanced along evolutionary trajectories through the accumulation of remote substitutions that alter conformational dynamics (Bhabha et al., 2013; Campbell et al., 2016; Curado-Carballada et al., 2019; Kaltenbach et al., 2018; Tokuriki and Tawfik, 2009; Vu et al., 2018). Modified protein dynamics are not mutually exclusive with the well-established concept of evolution of protein function *via* specific changes in shape or electrostatic character of active/binding sites, but serve as a complementary mechanism that may often overlap with these other more easily observable changes.”

REFERENCES REFERRED TO IN RESPONSE:

- Abdelkader EH, Feintuch A, Yao X, Adams LA, Aurelio L, Graham B, Goldfarb D, Otting G. 2015. Protein conformation by EPR spectroscopy using gadolinium tags clicked to genetically encoded *p*-azido-L-phenylalanine. *Chem Commun* **51**:15898–15901. doi:10.1039/c5cc07121f
- Bhabha G, Ekiert DC, Jennewein M, Zmasek CM, Tuttle LM, Kroon G, Dyson HJ, Godzik A, Wilson I a, Wright PE. 2013. Divergent evolution of protein conformational dynamics in dihydrofolate reductase. *Nat Struct Mol Biol* **20**:1243–9. doi:10.1038/nsmb.2676
- Campbell E, Kaltenbach M, Correy GJ, Carr PD, Porebski BT, Livingstone EK, Afriat-Jurnou L, Buckle AM, Weik M, Hollfelder F, Tokuriki N, Jackson CJ. 2016. The role of protein dynamics in the evolution of new enzyme function. *Nat Chem Biol* **12**:944–950. doi:10.1038/nchembio.2175
- Clifton BE, Jackson CJ. 2016. Ancestral protein reconstruction yields insights into adaptive evolution of binding specificity in solute-binding protein. *Cell Chem Biol* **23**:236–245. doi:10.1016/j.chembiol.2015.12.010
- Clifton BE, Kaczmarek JA, Carr PD, Gerth ML, Tokuriki N, Jackson CJ. 2018. Evolution of

- cyclohexadienyl dehydratase from an ancestral solute-binding protein. *Nat Chem Biol* **14**:542–547. doi:10.1038/s41589-018-0043-2
- Curado-Carballada C, Feixas F, Iglesias-Fernández J, Osuna S. 2019. Hidden conformations in *Aspergillus niger* monoamine oxidase are key for catalytic efficiency. *Angew Chemie Int Ed* **58**:3097–3101. doi:10.1002/anie.201812532
- Diederichs K, Karplus PA. 1997. Improved R-factors for diffraction data analysis in macromolecular crystallography. *Nat Struct Biol* **4**:269–275. doi:10.1038/nsbo497-269
- Fraser NJ, Liu JW, Mabbitt PD, Correy GJ, Coppin CW, Lethier M, Perugini MA, Murphy JM, Oakeshott JG, Weik M, Jackson CJ. 2016. Evolution of protein quaternary structure in response to selective pressure for increased thermostability. *J Mol Biol* **428**:2359–2371. doi:10.1016/j.jmb.2016.03.014
- Kaltenbach M, Burke JR, Dindo M, Pabis A, Munsberg FS, Rabin A, Kamerlin SCL, Noel JP, Tawfik DS. 2018. Evolution of chalcone isomerase from a noncatalytic ancestor. *Nat Chem Biol* **14**:548–555. doi:10.1038/s41589-018-0042-3
- Karplus PA, Diederichs K. 2012. Linking crystallographic model and data quality. *Science (80-)* **336**:1030 LP – 1033. doi:10.1126/science.1218231
- Lane LB. 1925. Freezing points of glycerol and its aqueous solutions. *Ind Eng Chem* **17**:924. doi:10.1021/ie50189a017
- Roos K, Wu C, Damm W, Reboul M, Stevenson JM, Lu C, Dahlgren MK, Mondal S, Chen W, Wang L, Abel R, Friesner RA, Harder ED. 2019. OPLS3e: Extending force field coverage for drug-like small molecules. *J Chem Theory Comput* **15**:1863–1874. doi:10.1021/acs.jctc.8b01026
- Ruggiero A, Dattelbaum JD, Staiano M, Berisio R, D’Auria S, Vitagliano L. 2014. A loose domain swapping organization confers a remarkable stability to the dimeric structure of the arginine binding protein from *Thermotoga maritima*. *PLOS One* **9**:e96560. doi:10.1371/JOURNAL.PONE.0096560
- Tokuriki N, Tawfik DS. 2009. Protein dynamism and evolvability. *Science (80-)* **324**:203–207. doi:10.1126/science.1169375
- Vu PJ, Yao XQ, Momin M, Hamelberg D. 2018. Unraveling allosteric mechanisms of enzymatic catalysis with an evolutionary analysis of residue-residue contact dynamical changes. *ACS Catal* **8**:2375–2384. doi:10.1021/acscatal.7b04263

REVIEWER COMMENTS

Reviewer #1 (Remarks to the Author):

While there was some clarification you did not fully address all of my comments and some of your responses raise further issues which need addressing before this work can be recommended for publication.

1. The newly added panels C-E in Fig 2 apparently correlate the modeled and experimental distances regardless of the dihedrals picked for the labels. What is the conclusion of better correlation for $\chi^2=180$ based on? By eye, there is no quantitative measure of correlation given. The statement in the caption that distributions broaden but peaks remain unchanged is confusing. Are you only correlating the most probable distributions (i.e., maxima) or distribution means? Common practice among practitioners is to overlay modeled and experimental distance distributions.
2. Your response that blips are now clearly mentioned in the captions appears inconsistent as you have removed the affected part of the data in fig S4 and identified them in the caption of S6.
3. I appreciate that you revalidated your data with broader parameters and updated fig 3 accordingly. Nevertheless, not updating Figure 3 with confidence intervals and reliability ranges is insufficient. These are important for the non-expert readers to understand the robustness of the distance data. By only displaying this information in the supporting figures you miss the point that those who would benefit most from this guidance are the least likely to deeply scrutinize the supporting EPR data.
4. "only the maxima of the mean distance distributions were considered" This statement is confusing. It is unclear whether you use the maxima or the means of your distance distributions. If the shape is unreliable the maximum will be unreliable too as it is defined by the distribution shape. The mean will be over the entire distribution and not over an arbitrarily chosen peak in a region where the shape is unreliable (as a "peak" is not clearly defined in such regions). This will likely worsen the apparent agreement of AncCDT-3/P188 with modelling.
5. The glass transition temperature of 20% water glycerol is significantly below the macroscopic freezing point. You appear to have a misconception about the relation between freezing point, glass transition temperature, and molecular dynamics upon shock freezing. The freezing point refers to crystallization. You have added cryoprotectant and cooled fast to avoid this. If it happened nevertheless, your sample is in a rather undefined state with respect to local glycerol concentration near the still solubilized protein. Given your freezing conditions, it is more likely that you ended up with a good glass. If that is so, the nominal freezing point of -6 C is of absolutely no consequence for what happened during shock-freezing of your sample. Crystallization requires nucleation. Shock freezing is intended to reach the glass transition temperature much faster than nucleation occurs and thus to avoid any significant crystallization. If your samples look transparent immediately after you take them out of the spectrometer, this has worked out. If so, no sudden change in dynamics has occurred at the freezing point. Viscosity changes smoothly, even across the glass transition temperature (Williams-Landel-Ferry equation). Near the glass transition temperature rotational and translational diffusion rates become much slower than most other relevant dynamic processes. This is where the sample solidifies, not at the freezing point. This much lower temperature will influence both binding equilibria and conformational equilibria.
6. Observing a substrate-induced conformational change in SeLAOBP is not exactly a replicate of AncCDT-1 +/- L-Arg. The question arises that if you made not replicates of this sample the only conformational change you observe could not be distinguished from any issue with sample quality that occasionally occur even in the best laboratories.
7. It is interesting that you refrain from reproducing mass spectrometry data used for estimating tagging yields.

Minor

Please check the red and black lines in Fig S4 B do not swap from left to middle panel. They do not according to caption but the minor blip at the end appears to go from the red to the black line.

Reviewer #2 (Remarks to the Author):

The authors have addressed my concerns.

Response to reviewer 2:

Specific points:

1. The newly added panels C-E in Fig 2 apparently correlate the modeled and experimental distances regardless of the dihedrals picked for the labels. What is the conclusion of better correlation for $\chi_6=180$ based on? By eye, there is no quantitative measure of correlation given. The statement in the caption that distributions broaden but peaks remain unchanged is confusing. Are you only correlating the most probable distributions (i.e., maxima) or distribution means? Common practice among practitioners is to overlay modeled and experimental distance distributions.

We understand the reviewer refers to the new Fig. S3 C-E of the Supporting Information. As we wrote in the figure legend, "The experimental distances correspond to the maxima of the respective DEER distance distributions." The same approach was used for the modeled distance distributions. This is now stated in the figure legend ("*The distances correspond to the maxima of the respective experimental or modelled DEER distance distributions.*") In addition, we clarified the sentence "Although the variation of these dihedral angles broadened the modeled distance distributions, the peaks of the Gd(III)–Gd(III) distance distributions changed only a little." to "Although the variation of these dihedral angles broadened the modelled distance distributions, the maxima of the Gd(III)–Gd(III) distance distributions did not change by more than 0.7 nm." We do not show overlays between modeled and experimental distance distributions, as we consider the modeling to be too coarse to reproduce distance distribution widths in a meaningful way. It is perfectly possible that $\chi_6=180$ is not the best choice for modeling. We do not have enough data points in Fig. S3C and D to draw statistically significant conclusions and, hence, did not calculate correlation coefficients. The correlation in Fig. S3D appears better than the correlation in Fig. S3C and therefore we chose $\chi_6=180$. When we allowed variations of additional dihedral angles in the AzF-propargyl-DO3A-Gd residue, the correlation seemed to worsen again (Fig. S3E versus Fig. S3D), but this seemed to apply mainly to two of the orange data points. Again, we believe that the number of data points is too small to draw conclusions beyond providing an impression of the reliability of the comparison of the experimental distances with modeled distances.

2. Your response that blips are now clearly mentioned in the captions appears inconsistent as you have removed the affected part of the data in fig S4 and identified them in the caption of S6.

We removed the sentence from Fig. S4 and left it in Fig. S6 as in B there is still a small artefact at the end which was not removed.

3. I appreciate that you revalidated your data with broader parameters and updated fig 3 accordingly. Nevertheless, not updating Figure 3 with confidence intervals and reliability ranges is insufficient. These are important for the non-expert readers to understand the robustness of the distance data. By only displaying this information in

the supporting figures you miss the point that those who would benefit most from this guidance are the least likely to deeply scrutinize the supporting EPR data.

We added the uncertainty range and reliability ranges to the figure as the referee requests.

4. “only the maxima of the mean distance distributions were considered” This statement is confusing. It is unclear whether you use the maxima or the means of your distance distributions. If the shape is unreliable the maximum will be unreliable too as it is defined by the distribution shape. The mean will be over the entire distribution and not over an arbitrarily chosen peak in a region where the shape is unreliable (as a “peak” is not clearly defined in such regions). This will likely worsen the apparent agreement of AncCDT-3/P188 with modelling.

We concede that the term “mean distance distribution” may have been misleading. The statement in our previous response letter was “We did not use or interpret the shape of the distance distributions of these variants anywhere in our conclusions, i.e. only the maxima of the mean distance distributions were considered (which are reliable).” To clarify, this statement, which did not appear in the main text or supporting information, does not refer to any specific calculation of mean values, as we merely took the maxima of the distance distributions. We maintain that the positions of maxima are more reliable than other points of the distance distributions.

5. The glass transition temperature of 20% water glycerol is significantly below the macroscopic freezing point. You appear to have a misconception about the relation between freezing point, glass transition temperature, and molecular dynamics upon shock freezing. The freezing point refers to crystallization. You have added cryoprotectant and cooled fast to avoid this. If it happened nevertheless, your sample is in a rather undefined state with respect to local glycerol concentration near the still solubilized protein. Given your freezing conditions, it is more likely that you ended up with a good glass. If that is so, the nominal freezing point of -6 C is of absolutely no consequence for what happened during shock-freezing of your sample. Crystallization requires nucleation. Shock freezing is intended to reach the glass transition temperature much faster than nucleation occurs and thus to avoid any significant crystallization. If your samples look transparent immediately after you take them out of the spectrometer, this has worked out. If so, no sudden change in dynamics has occurred at the freezing point. Viscosity changes smoothly, even across the glass transition temperature (Williams-Landel-Ferry equation). Near the glass transition temperature rotational and translational diffusion rates become much slower than most other relevant dynamic processes. This is where the sample solidifies, not at the freezing point. This much lower temperature will influence both binding equilibria and conformational equilibria.

Glycerol is a commonly used additive to prevent crystallization upon freezing. Crystallization of the water affects the homogenous distribution of the proteins in the solution and causes islands of high protein concentration. This affects the background decay and the phase memory time. We freeze the samples by either immersing into liquid nitrogen or by inserting the sample into a precooled sample position (10-50K) in

the magnet. This cooling procedure has been referred to as slow freezing (~1 sec) by Freed and co-workers. Their article “Effect of freezing conditions on distances and their distributions derived from Double Electron Electron Resonance (DEER): A study of doubly-spin-labelled T4 lysozyme” <https://doi.org/10.1016/j.jmr.2012.01.004> reports that the freezing temperature of 10 % glycerol in water placed in an EPR tube and dipped into liquid nitrogen is -3 °C, so the freezing point of 20 % glycerol in water should be somewhat lower. The main effect of glycerol is in lowering the background decay rather than changing the distance distribution. The slowest background decay is achieved with rapid freeze quench (see Figures 3 and 4 of the paper mentioned above) and there a change is observed in the distance distribution. As our samples belong to the slow freezing category according to Freed’s paper we do not have a perfect glass, but a good enough glass to have a sufficiently long phase memory time and a moderate background decay. More glycerol would bring us closer to a perfect glass but the reduced water content would be less physiological. We estimate the freezing temperature to be about -6 °C but protein diffusion would become negligible already before the freezing point. We do not expect dramatic shifts in conformational and binding equilibria, even if the actual freezing temperature would have been lower than -6 °C. The method of sample freezing was stated on page 25.

6. Observing a substrate-induced conformational change in *Se*LAOBP is not exactly a replicate of AncCDT-1 -/+ L-Arg. The question arises that if you made not replicates of this sample the only conformational change you observe could not be distinguished from any issue with sample quality that occasionally occur even in the best laboratories.

We repeated the sample preparation, starting from a new batch of *E. coli* cells, and reproduced the bimodal distance distribution of refolded ligand-free AncCDT-1. We have added the following panels to Supplementary Figure 4I.

7. It is interesting that you refrain from reproducing mass spectrometry data used for estimating tagging yields.

We have attached the mass spectrometry data of all proteins (for review only). In most of the samples, there was no sign of incomplete tagging. We chose not to include those data in the supporting information because mass spectrometry is an unreliable method for quantifying protein amounts whereas the DEER data prove that double-tagging was achieved successfully. We changed the corresponding sentence on page 9 of the main

text to clarify further: *"...and mass spectrometry analysis indicated that the tag ligation yields were sufficient to deliver at least 75 % doubly tagged samples, except for the AncCDT-1 mutant at sites 138 and 161, where at least 25 % of the sample was doubly tagged."*

Minor

Please check the red and black lines in Fig S4 B do not swap from left to middle panel. They do not according to caption but the minor blip at the end appears to go from the red to the black line.

Thank you, this was fixed.

REVIEWER COMMENTS

Reviewer #1 (Remarks to the Author):

The authors have addressed most of my concerns and I do believe the general conclusions are supported by the data presented. I do, however, take issue with some lack of rigor in data analysis. As this manuscript will undoubtedly serve as an example to younger or less experienced scientists I urge the authors to consider the following points.

1. Why do you apply double measures to the inclusion of data. You show the MS data to the stubborn reviewer but do not want to include it. Either the data is good enough to support your conclusions or not. Asking for this data has already changed quantitative statements in your manuscript and research transparency would mandate this to be shown for the interested reader to make their own judgement. Why would this be different for your case or this data? The other way around, if you were not happy the quality is publishable, should you be conclusions on it? Double tagging is evident in DEER so the MS is merely confirming the obvious and potentially quantifying it. If you were unhappy with its quality it would not be essential. Nevertheless if it is interpreted it must be shown.
2. The maxima of peaks in a distance probability distribution are only well defined if the peaks themselves are well defined. The issue is that peaks towards the longer distance range become less well defined. So you do not know if there are two peaks or one with the maximum between the two peaks (i.e. not coinciding with their maxima). Thus, I disagree with you statement maxima were more reliable. If there was data/studies showing this you would have referenced them. The developer of the software you use for data analysis clearly states that for the long distance region you interpret the shape of the distribution cannot be reliably defined but only its mean and potentially its width. This means as long as you do not have longer data you cannot interpret the peaks (=shape) unless you can prove the chair of chemistry at ETH wrong on his own software. An endeavour I would not dare to undertake. On the other hand even just interpreting mean and width the change is obvious. The nice (coincidental) numerical agreement might be lost but that is not essential for your conclusions.
3. Freed's paper also shows supercooling to -12C when cooling very slowly rather than quickly while vigorously shaking the sample. As you freeze quickly you should even reach lower temperatures before crystallisation sets in. I doubt you can vigorously shake your W-band capillaries to speed up nucleation. With a higher percentage of cryoprotectant and faster freezing you will approach the glass transition temperature much closer rather than the macroscopic freezing point when the proteins immobilise. The fact that you do not know the temperature the binding equilibrium freezes out and what the temperature dependence of your binding equilibrium is makes the agreement coincidental and is a red herring. That you stress you do not expect a change in binding with temperature sounds almost like you were justifying a scientific observation with naivety. Meeting even well founded expectations has not always been the best guide to scientific discovery. Not having an expectation is unlikely to be a better guide.

For your manuscript to (rightly) become a landmark paper that serves as a formidable example to the next generation of practitioners I would expect these three points to be addressed/discussed openly in the final manuscript version. At this stage I would consider it one of many examples where scientific rigor came second to getting a smooth story published with the technical issues glossed over.

1. Why do you apply double measures to the inclusion of data. You show the MS data to the stubborn reviewer but do not want to include it. Either the data is good enough to support your conclusions or not. Asking for this data has already changed quantitative statements in your manuscript and research transparency would mandate this to be shown for the interested reader to make their own judgement. Why would this be different for your case or this data? The other way around, if you were not happy the quality is publishable, should you be conclusions on it? Double tagging is evident in DEER so the MS is merely confirming the obvious and potentially quantifying it. If you were unhappy with its quality it would not be essential. Nevertheless if it is interpreted it must be shown.

We have included the mass spectrometry data in Supplementary Figure 4. The figure legend of Supplementary Figure 4 reads:

“Supplementary Figure 4. Mass spectra of protein samples before and after tagging. The left and right panels show the mass spectra before and after tagging, respectively, together with the expected masses (red). Stars identify mass peaks attributed to singly tagged protein. (A) Refolded AncCDT-1, AzF in positions 68 and 138. (B) Refolded AncCDT-1, AzF in positions 138 and 161. (C) AncCDT-1, AzF in positions 68 and 219. (D) AncCDT-3, AzF in positions 68 and 138. (E) AncCDT-5, AzF in positions 68 and 138. (F) PaCDT, AzF in positions 68 and 139.”

2. The maxima of peaks in a distance probability distribution are only well defined if the peaks themselves are well defined. The issue is that peaks towards the longer distance range become less well defined. So you do not know if there are two peaks or one with the maximum between the two peaks (i.e. not coinciding with their maxima). Thus, I disagree with your statement maxima were more reliable. If there was data/studies showing this you would have referenced them. The developer of the software you use for data analysis clearly states that for the long distance region you interpret the shape of the distribution cannot be reliably defined but only its mean and potentially its width. This means as long as you do not have longer data you cannot interpret the peaks (=shape) unless you can prove the chair of chemistry at ETH wrong on his own software. An endeavour I would not dare to undertake. On the other hand even just interpreting mean and width the change is obvious. The nice (coincidental) numerical agreement might be lost but that is not essential for your conclusions.

Although the validation procedure that we carried out gives the confidence interval of the distance distribution and we think that this is sufficient (see for example a recent publication in Nat. Comm. (NATURE COMMUNICATIONS | (2019) 10:4619 | <https://doi.org/10.1038/s41467-019-12591-x>, where the same was applied) we did follow the reviewer's request. We added on p. 14:

“Although the confidence ranges identify clear peak maxima, we also calculated mean distance distributions (caption of **Supplementary Figure 7**) to further assess the reliability of these peak positions due to the limited evolution time with respect to the long distance observed. This resulted in shifts of at most about 0.25 nm.”

And in the caption of Supplementary Figure 7:

“As the main peak in each DEER distance distribution appears in the orange region, we also calculated means of the distance distributions to assess potential contributions of the smaller peaks nearby. We considered two ranges for the calculation, where the first range (4.0–6.5 nm) included the small peak appearing in the yellow region next to the main peak while the second range included only the most intense peak (5.0–6.5 nm). The mean distance values obtained were: (A) native: 4.96 ± 0.42 and 5.18 ± 0.26 nm (original peak maximum at 5.2 nm); (A) refolded: 5.15 ± 0.57 and 5.44 ± 0.24 nm (original peak maximum at 5.4 nm); (B) native: 5.26 ± 0.41 nm and 5.47 ± 0.20 nm (original peak maximum at 5.4 nm); (B) refolded: 5.62 ± 0.24 (original peak maximum at 5.7 nm).”

3. Freed's paper also shows supercooling to -12C when cooling very slowly rather than quickly while vigorously shaking the sample. As you freeze quickly you should even reach lower temperatures before crystallisation sets in. I doubt you can vigorously shake your W-band capillaries to speed up nucleation. With a higher percentage of cryoprotectant and faster freezing you will approach the glass transition temperature much closer rather than the macroscopic freezing point when the proteins immobilise. The fact that you do not know the temperature the binding equilibrium freezes out and what the temperature dependence of your binding equilibrium is makes the agreement coincidental and is a red herring. That you stress you do not expect a change in binding with temperature sounds almost like you were justifying a scientific observation with naivety. Meeting even well founded expectations has not always been the best guide to scientific discovery. Not having an expectation is unlikely to be a better guide.

We changed the sentence on p. 22 as follows.

“Considering that DEER data are recorded of frozen solutions, it cannot be ruled out that conformational equilibria established at room temperature change during snap-freezing of the samples. In principle, the freezing point of 20 % deuterated glycerol solutions is at most 10 degrees below 0 °C,⁶⁷ but the actual temperature at which the dynamics stopped is difficult to determine. As all samples were frozen in the same way, however, the conformational differences observed between the samples refer to similar conditions.”